# AdaGrad Converges in a Robust Sense: Almost Sure Last-Iterate Rates under Any Stopping Time

## Abstract

AdaGrad has become a widely used algorithm for training deep models. Recently, the study of almost sure last-iterate convergence rates in stochastic optimization has attracted increasing attention, as it provides the guarantee of stability and **robustness** for arbitrary single trajectory. While such results are well understood for stochastic gradient descent (SGD), the corresponding analysis for AdaGrad remains limited. In this paper, we establish **almost sure** convergence rates of AdaGrad for the **last-iterate** in the (strongly) convex setting and for the best-iterate in the non-convex setting, both valid under **arbitrary** stopping times and with a flexible dependence on gradient history.

## 1 Introduction

Adaptive gradient methods in broad, such as AdaGrad (Duchi et al., 2011) Adam (Kingma & Ba, 2014), and AdamW (Loshchilov & Hutter, 2019), have attracted much attention from the machine learning community due to their efficiency in training deep neural networks and large language models (Devlin et al., 2019; Touvron et al., 2023). Because of the widespread empirical success, understanding the theoretical convergence properties of AdaGrad (and its variants) has become an active research area that is critical for both machine learning and optimization communities. However, in contrast to extensive studies on convergence properties for stochastic gradient descent (SGD), AdaGrad remains less explored in depth and width.

Existing works on the convergence of AdaGrad focus mainly on the convergence rates in expectation or convergence rates with high probability (*w.h.p.*), e.g., Wang et al. (2023) and Attia & Koren (2023), which potentially provides a probability dependence convergence guarantee for a single trajectory of training process. However, incorporating probability parameter $\delta$ into convergence rate often is less favored compared to so called *almost sure* convergence, which guarantees the convergence rate in probability 1. Recent works of Sebbouh et al. (2021) and Liu & Yuan (2022) establish the almost sure convergence rates for stochastic gradient descent (SGD) and its variants (Stochastic heavy-ball and Nesterov Acceleration Gradient) respectively. To the best of our knowledge, there is no a satisfactory answer to the almost sure convergence rates of AdaGrad.

### 1.1 Motivation

In this subsection, we will discuss unsolved theoretical issues in existing research on AdaGrad, together with practical benefits of varying the power of gradient in adaptive step-sizes. Generally speaking, in the theoretical side, the question—obtaining the **last-iterate** almost sure convergence rates in the (strongly) convex case and best-iterate almost sure convergence rates in the non-convex case for **any time** $t \geq 1$—remains open; in the practical side, providing more flexibility for practitioners to implement AdaGrad is relatively less explored.

#### 1.1.1 Almost Sure Last Iterate Convergence

Before formal discussion on the theoretical issue for AdaGrad, we clarify and highlight the difference between convergence rates with high probability (with **polylog**$(\frac{1}{\delta})$ term) and almost sure (*a.s.*)

convergence rates, as well as the difference between last-iterate convergence and average-iterate convergence.

**Convergence a.s. vs Convergence w.h.p.** Although convergence rates with high probability have been the prevailing standard in stochastic optimization, almost sure convergence rates provide a strictly stronger and more robust guarantee. High-probability bounds ensure good performance with confidence $1 - \delta$ at a fixed time horizon, but they always leave open the possibility of failure events that may occur, and their guarantees typically degrade under adaptive choices such as any random stopping times $t \geq 1$. In contrast, almost sure convergence rates rule out such failures on all but a measure-zero set of trajectories, ensuring that along almost every realization the algorithm's iterates stabilize at the claimed rate. This trajectory-wise stability is particularly important in machine learning, where one typically observes and deploys a single run of the algorithm, not an ensemble. Moreover, almost sure results guarantee robustness to any stopping time $t \geq 1$ and other adaptive procedures that rely on the evolution of a single trajectory. Thus, while high-probability rates offer finite-horizon confidence, almost sure convergence rates capture the trajectory-level stability in any time $t \geq 1$ that is essential for both theoretical completeness and practical reliability.

**Last-iterate vs Average-iterate convergence.** While average-iterate convergence has long served as the standard benchmark in stochastic optimization—owing to its analytical tractability and its ability to reduce variance—last-iterate convergence is of greater practical importance. In modern machine learning applications, the model used in deployment is almost always the final iterate rather than an averaged solution. Moreover, averaging is often memory consuming in practice, as it requires storing or recombining all past iterates, and in nonconvex settings it may even obscure the true behavior of the optimization trajectory. By contrast, last-iterate guarantees directly reflect the stability and robustness of the actual optimization trajectory. They are particularly critical when training procedures are stopped adaptively, for example through early stopping or validation-based criteria, where only the current iterate matters. For these reasons, establishing last-iterate convergence rates not only deepens our theoretical understanding of stochastic optimization dynamics but also ensures that theoretical guarantees align closely with practical usage.

The foregoing analysis reveals two unresolved issues in the theoretical understanding of AdaGrad:

- **Issue 1:** To the best of our knowledge, almost sure convergence rates for AdaGrad for **any time** $t \geq 1$ have not been established in the existing literature.

- **Issue 2:** For the convex case, even in the sense of expectation, the **last-iterate** convergence rates of AdaGrad remain unaddressed in existing works.

### 1.1.2 PRACTICAL BENEFITS OF FLEXIBILITY

Before formal discussion on the practical benefits of introducing flexibility parameter, we recall the original AdaGrad: [1]

$$
\begin{aligned}
x_{t+1} &= x_t - \eta_t\, g_t, \\
\eta_t &= \frac{a}{\left(b + \sum_{i=1}^{t} \|g_i\|^2\right)^{\frac{1}{2}}},
\end{aligned} \tag{AdaGrad-Norm}
$$

where $a > 0$, $b > 0$ and $\{g_i\}_{i=1}^{t}$ are the historical stochastic gradients.

It is worth noting that Li & Orabona (2019) introduced a new parameter $\epsilon \in [0, \frac{1}{2}]$ in the denominator, which improves the decay rate of the stepsize. If the new parameter $\epsilon$ is incorporated into the stepsize used in (AdaGrad-Norm), the stepsizes turn into

$$
\eta_t = \frac{a}{\left(b + \sum_{i=1}^{t} \|g_i\|^2\right)^{\frac{1}{2}+\epsilon}}.
$$

Furthermore, extensive experiments in Choudhury et al. (2024) demonstrated that AdaGrad with $\epsilon = \frac{1}{2}$ outperforms the original (AdaGrad-Norm) with $\epsilon = 0$. These observations suggest that

---

[1]We clarify that we only discuss NORM adaption in this paper, other adaptions including DIAG and FULL matrix in Duchi et al. (2011) are beyond the scope of this paper.

adjusting the scale parameter $\frac{1}{2}$ in (AdaGrad-Norm) may lead to better empirical performance. Motivated by this, if we adjust the exponent 2 on $\|g_i\|$, another scale parameter in (AdaGrad-Norm), with a tunable flexibility parameter $\gamma \in [0, 2]$ such that

$$\eta_t = \frac{a}{\left(b + \sum_{i=1}^{t} \|g_i\|^{\gamma}\right)^{\frac{1}{2}+\epsilon}},$$

we investigate whether replacing the fixed exponent 2 on $\|g_i\|$ with $\gamma \in [0, 2]$ can further enhance performance. We answer this question affirmatively through experiments on training the VGG+BN+Dropout network in Wilson et al. (2017) on the CIFAR-10 dataset (see Figure 1), where we find that choices such as $\gamma = 0.1$ and $\gamma = 1$ yield better performance than the standard setting $\gamma = 2$. See Appendix A for more details of experiments.

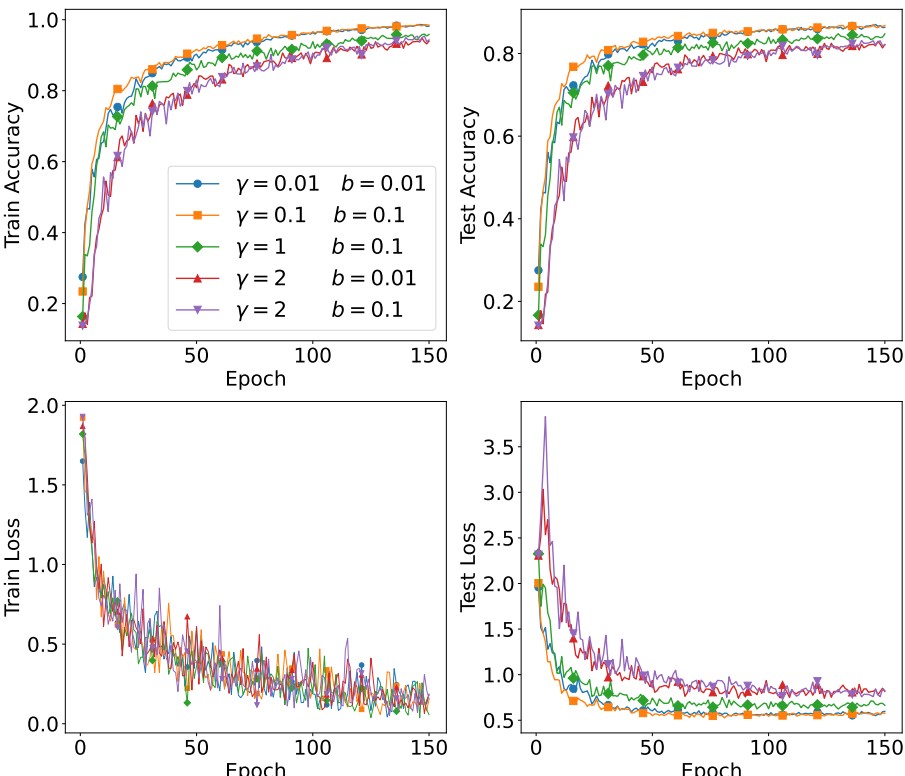

Figure 1: The training and testing accuracy/loss on CIFAR-10 using AdaGrad with varying $\gamma$ and $b$. **Case (i):** when $b = 0.1$, the performance of $\gamma = 0.1$ and $\gamma = 1$ outperform $\gamma = 2$; **Case (ii):** when $b = 0.01$, the performance $\gamma = 0.1$ outperforms $\gamma = 2$. The configuration $\gamma = 2$ shows consistently inferior performance in both cases. See Appendix A for more discussions on the results of experiments.

Motivated by the above limitations and benefits, we may naturally ask the following **question:**

*Can we establish **last-iterate**/best-iterate **almost sure** convergence rates for AdaGrad with flexibility in the (strongly) convex/non-convex cases for **any time** $t \geq 1$?*

It should be pointed out that initiated by Li & Orabona (2019), the stepsize of (AdaGrad-Norm) has two forms depending on whether the current gradient oracle $g_t$ is included in the denominator of $\eta_t$. For the sake of completeness and generality, in this paper, we will consider a general variant of (AdaGrad-Norm), named as (FlexAdaGrad-Norm), with both types of stepsizes (Type I) and (Type II) which are listed in the following:

$$x_{t+1} = x_t - \eta_t\, g_t \qquad \text{(FlexAdaGrad-Norm)}$$

where

$$\eta_t = \frac{a}{\left(b + \sum_{i=1}^{t-1} \|g_i\|^\gamma\right)^{\frac{1}{2}+\epsilon}} \tag{Type I}$$

and

$$\eta_t = \frac{a}{\left(b + \sum_{i=1}^{t} \|g_i\|^\gamma\right)^{\frac{1}{2}+\epsilon}}, \tag{Type II}$$

where $b > 0$, $\epsilon \in (0, \frac{1}{2}]$, and $\gamma \in (0, 2]$ is the *flexibility parameter* quantifying the contribution of gradient history to the adaptive stepsize. As an end of this subsection, we remark that with loss of generality, we can actually set $a = 1$ for theoretical analysis below.

## 1.2 OUR CONTRIBUTIONS

To address the question proposed in Section 1.1, we make the following contributions in this paper:

(1) We introduce a flexibility parameter $\gamma$ to control the impact of gradient history in the original (AdaGrad-Norm), thereby providing more flexibility for the practitioners compared to original (AdaGrad-Norm);

(2) We establish the **last-iterate** almost sure convergence rates in **any time** $t \geq 1$ for (FlexAdaGrad-Norm) under stepsizes Type I and Type II in both strongly convex and convex cases;

(3) We provide the best-iterate almost sure convergence rates in any time $t \geq 1$ for (FlexAdaGrad-Norm) under stepsizes Type I and Type II in the non-convex case;

The main results of this paper can be summarized below (also refers to Table 1).

**Theorem 1.1** (Informal). *Suppose the adaptive stepsize $\eta_t$ takes the form of (Type I) or (Type II). Then the following almost sure convergence rates hold for (FlexAdaGrad-Norm).*

- *If $\gamma \in (0, 2)$, the last iterate convergence rate in any time $t \geq 1$ of (FlexAdaGrad-Norm) is $o\left(\frac{1}{t^{1-\eta}}\right)$ for strongly convex functions, and $O\left(\frac{1}{t^{\frac{1}{2}-\epsilon}}\right)$ for convex functions.*

- *If $\gamma \in (0, 2]$, the best iterate convergence rate in any time $t \geq 1$ of (FlexAdaGrad-Norm) is also $O\left(\frac{1}{t^{\frac{1}{2}-\epsilon}}\right)$ for nonconvex functions.*

## 1.3 RELATED WORKS

**Almost sure convergence.** Since Robbins & Monro (1951) constructed a stochastic approximation algorithm in the 1950s, the almost sure convergence of stochastic algorithms became an important topic (See Pemantle (1990); Benaim & Hirsch (1995); Benaïm (2006); Mertikopoulos et al. (2020)). Subsequently, Robbins & Siegmund (1971) provides an important result (a.k.a Robbins-Siegmund Theorem), which has been the cornerstone of analyzing the almost sure convergence. Bertsekas & Tsitsiklis (2000) obtained that stochastic gradient descent (SGD) converges asymptotically to critical points with probability 1 via complicated analysis. Orabona (2020a) simplified the proof of Bertsekas & Tsitsiklis (2000) via an elegant series result. For obtaining a non-asymptotical convergence with probability 1, Pelletier (1998); Godichon-Baggioni (2019) got the almost sure convergence rates for locally strong convex functions. Sebbouh et al. (2021) employed the IMA trick to SGD to get the almost sure convergence rates in the general convex case. Liu & Yuan (2022) obtained the almost sure convergence rates for SGD without IMA in strongly convex, convex and nonconvex cases. Karandikar & Vidyasagar (2024) also extended the almost sure convergence rates to functions satisfying PL condition. Besides, there are two variants of SGD including stochastic heavy ball (SHB) and stochastic Nesterov acceleration gradient (SNAG) in existing literature. Almost sure convergence rates for SHB and SNAG can be found in Sebbouh et al. (2021); Liu & Yuan (2022).

| Algorithm | Stepsize | $f$ | Flex. | Rates | Iterate | Stop time |
|---|---|---|---|---|---|---|
| SGD (Sebbouh et al. (2021)) | diminish | C | N/A | $o\left(\frac{1}{t^{\frac{1}{2}-\epsilon}}\right)$ | average | any time |
| SGD (Liu & Yuan (2022)) | diminish | SC | N/A | $o\left(\frac{1}{t^{1-\epsilon}}\right)$ | last | any time |
| SGD (Liu & Yuan (2022)) | diminish | C | N/A | $O\left(\frac{1}{t^{\frac{1}{3}-\epsilon}}\right)$ | last | any time |
| SGD (Liu & Yuan (2022)) | diminish | NC | N/A | $O\left(\frac{1}{t^{\frac{1}{2}-\epsilon}}\right)$ | best | any time |
| AdaGrad(Attia & Koren (2023)) | Type II | C | $\gamma = 2$ | $O\left(\frac{\sqrt{\log t}}{\sqrt{t}}\right)$ | average | $t \geq t_0, t_0$ unknown |
| AdaGrad(Attia & Koren (2023)) | Type II | NC | $\gamma = 2$ | $O\left(\frac{\log^2 t}{\sqrt{t}}\right)$ | best | $t \geq t_0, t_0$ unknown |
| **AdaGrad (Ours)** | Type I, II | SC | $\gamma \in (0,2)$ | $o\left(\frac{1}{t^{1-\eta}}\right)$ | **last** | any time |
| **AdaGrad (Ours)** | Type I, II | C | $\gamma \in (0,2)$ | $O\left(\frac{1}{t^{\frac{1}{2}-\epsilon}}\right)$ | **last** | any time |
| **AdaGrad (Ours)** | Type I, II | NC | $\gamma \in (0,2]$ | $O\left(\frac{1}{t^{\frac{1}{2}-\epsilon}}\right)$ | best | any time |

Table 1: Comparisons on almost sure convergence rates. **"SC", "C" and "NC"** mean strongly convex, convex and nonconvex $f(x)$ respectively. **"diminish"** means the diminishing stepsizes satisfying $\sum_t \eta_t = \infty$, $\sum_t \eta_t^2 < \infty$. **"any time"** means the rates are valid in any stopping time $t \geq 1$. The almost sure convergence rates in the gray rows are natural derivation of Attia & Koren (2023), whose derivation can be found in Appendix D.

**Last-iterate convergence.** The research on last-iterate convergence of stochastic algorithms can date back the stochastic approximation method which proposed by Robbins and Monro Robbins & Monro (1951). For SGD, Zhang (2004) provided the last-iterate convergence of SGD in unbounded domains with constant stepsizes, while Shamir & Zhang (2013) showed that the last-iterate of SGD with diminishing stepsizes converges in bounded domains. Based on these two torks, Orabona (2020b) provided the last-iterate convergence of SGD in unbounded domains with diminishing stepsizes. Moving beyond asymptotic guarantees, Liu & Yuan (2022) provided non-asymptotic rates of SGD and its two variants. Besides SGD, Li & Orabona (2019) presented the last-iterate convergence of AdaGrad without non-asymptotic rates.

**AdaGrad.** Auer et al. (2002) and Duchi et al. (2011) made significant contributions to the early exploration of optimization methods with adaptive stepsizes. Most convergence guarantees for AdaGrad lie in the sense of expectation or high probability.Mukkamala & Hein (2017) and Reddi et al. (2018) investigated the convergence in the strongly convex case. Moving beyond convexity, Chen et al. (2018) and Zhou et al. (2018) investigated the convergence rates (in expectation) for AdaGrad in the nonconvex setting. Other research on the convergence (in expectation) for AdaGrad can refer to Wu et al. (2018); Leluc & Portier (2023); Faw et al. (2023); Wang et al. (2023) and reference therein. Attia & Koren (2023) researched the convergence rates with high probability for AdaGrad. Besides, Li & Orabona (2019) provided the asymptotically almost sure convergence to critical points with probability 1. Additionally, Choudhury et al. (2024) investigated a variant of AdaGrad that removes the square root in the denominator, which achieves convergence rates comparable to the original AdaGrad while demonstrating strong practical performance.

## 2 PRELIMINARY

**Notation.** For any two sequences $\{A_n\}$ and $\{B_n\}$, denote $A_n = O(B_n)$, if there exists $c > 0$ such that $\lim_{n\to\infty} \frac{A_n}{B_n} \leq c$; Denote $A_n = o(B_n)$, if $\lim_{n\to\infty} \frac{A_n}{B_n} = 0$.

## 2.1 PROBLEM SET UP

Throughout this paper, we consider the following stochastic optimization problem:

$$\min_{x \in \mathbb{R}^d} f(x), \tag{P}$$

where $f(x) \equiv \mathbb{E}_\xi[F(x, \xi)]$ and $\xi$ encodes the uncertainty from the environment. $f(x)$ is continuously differentiable. Without loss of generality, denote any minimizer by $x^* \in \arg\min_{x \in \mathbb{R}^d} f(x)$ and the minimum by $f^* = f(x^*)$. We denote the conditional expectation with respect to the past observation $\xi_1, \xi_2 \cdots, \xi_{t-1}$ by $\mathbb{E}_t[\cdot]$. Besides, we call $f(x)$ $\mu$-strongly convex, if for all $x, y \in \mathbb{R}^d$, there exists $\mu > 0$ such that

$$f(y) \geq f(x) + \langle \nabla f(x), y - x \rangle + \frac{\mu}{2}\|y - x\|^2.$$

In this paper, we make the following two assumptions on $f(x)$, which are standard and have been broadly used in former works on convergence analysis of stochastic optimization (see Chen et al. (2019); Kavis et al.; Liang et al. (2025)).

**Assumption 2.1.** $f(x)$ is $M$-smooth, i.e., for all $x, y \in \mathbb{R}^d$,

$$\|\nabla f(x) - \nabla f(y)\| \leq M\|x - y\|.$$

**Assumption 2.2.** The unbiased stochastic gradient oracle sequence $\{g_t\}_{t=1}^\infty$ is uniformly bounded, i.e., there exists $Q > 0$ such that

$$\|g_t\| \leq Q.$$

**Remark 2.3.** *Assumption 2.1 is well known to be useful in obtaining the descent inequality in broad literature. Note that the stochastic gradient oracle $g_i$ can be decomposed into the exact gradient $\nabla f(x_i)$ and noise oracle $u_i$ as follows:*

$$g_i = \nabla f(x_i) + u_i.$$

*Thus, Assumption 2.2 often has another form in existing literature on almost sure convergence of stochastic optimization (Li & Orabona, 2019; Mertikopoulos et al., 2020). In these works, the authors assume that there exists constants $L > 0$ and $\sigma > 0$ such that $\|\nabla f(x_i)\| \leq L$ and $\|u_i\| \leq \sigma$, which naturally implies Assumption 2.2. Technically, Assumption 2.2 ensure the stepsize satisfying $\sum_{t=1}^\infty \eta_t = \infty$ almost surely, which is a presumed condition in almost sure convergence analysis for SGD. We would like to point out that Assumption 2.2 is also crucial in obtaining so called "generalized square summability" for adaptive stepsizes, which generalizes the condition of $\sum_{t=1}^\infty \eta_t^2 < \infty$ in SGD. Detailed analysis is deferred to Section 3.1.*

The following Lemma 2.4, also known as Robbins-Siegmund Theorem, is a classical supermartingale convergence result that was proven in Robbins & Siegmund (1971). This lemma has been successfully leveraged to obtain almost sure convergence rates for SGD (see Liu & Yuan (2022); Sebbouh et al. (2021)), and will also be used in our analysis later.

**Lemma 2.4** (Robbins-Siegmund Theorem). *Let $\{X_t\}$, $\{Y_t\}$, and $\{Z_t\}$ be three sequences of random variables that are adapted to a filtration $\{\mathcal{F}_t\}$. Let $\{\theta_t\}$ be a sequence of nonnegative real numbers such that $\prod_{t=1}^\infty (1 + \theta_t) < \infty$. Suppose that the following conditions hold:*

*(1) $X_t, Y_t$, and $Z_t$ are nonnegative for all $t \geq 1$;*

*(2) $\mathbb{E}[Y_{t+1} \mid \mathcal{F}_t] \leq (1 + \theta_t) Y_t - X_t + Z_t$ for all $t \geq 1$;*

*(3) $\sum_{t=1}^\infty Z_t < \infty$ holds almost surely.*

*Then $\sum_{t=1}^\infty X_t < \infty$ almost surely and $Y_t$ converges almost surely.*

## 3 MAIN RESULTS

In Section 3.1, We will present a key proposition used in our proofs, which are our main technical contributions. Then we present the almost sure convergence rates of (FlexAdaGrad-Norm) with stepsizes (Type I) and (Type II) in Section 3.2 and 3.3.

### 3.1 GENERALIZED SQUARE SUMMABILITY

We initiated the discussion on the conditions that stepsize $\eta_t$ should satisfy in Remark 2.3. Before stating our main results, we will elaborate on condition necessary for almost sure convergence rate of (FlexAdaGrad-Norm). The subtlety of the inequality in the following proposition indicates the fundamental difference between the condition $\sum_{t=1}^{\infty} \eta_t^2 < \infty$ for SGD and its counterpart for (FlexAdaGrad-Norm). As we will see in this section that inequality (1) generalizes the square summability of the stepsizes of SGD, we call inequality (1) the *Generalized Square Summability*. This technical contribution regarding this point is formally stated as follows.

**Proposition 3.1.** *Suppose Assumption 2.2 hold. Let $b > 0$, $\beta > 0$, $0 < \eta < 1$ and $0 < \gamma < 2$. Set $\eta$ and $\gamma$ such that $2(1 - \eta) \le 1 - \frac{\gamma}{2}$. Then we have*

$$\sum_{t=1}^{\infty} \mathbb{E}_t \left[ t^{1-\eta} \frac{\|g_t\|^2}{\left(b + \sum_{i=1}^{t} \|g_i\|^\gamma\right)^{1+\beta}} \right] < \infty. \tag{1}$$

*In particular, if $\beta = 2\epsilon$, for (Type I) and (Type II), we have*

$$\sum_{t=1}^{\infty} \mathbb{E}_t \left[ t^{1-\eta} \eta_t^2 \|g_t\|^2 \right] \overset{Equ.(19)}{=} O \left( \sum_{t=1}^{\infty} \mathbb{E}_t \left[ t^{1-\eta} \frac{\|g_t\|^2}{\left(b + \sum_{i=1}^{t} \|g_i\|^\gamma\right)^{1+2\epsilon}} \right] \right) < \infty.$$

To see that above summability is a proper replacement of $\sum_{t=1}^{\infty} \eta_t^2 < \infty$ in SGD, one can choose $\eta_t = \Theta\left(\frac{1}{t^{1-\theta}}\right)$ for SGD, and we have

$$\sum_{t=1}^{\infty} \mathbb{E}_t \left[ t^{1-\eta} \eta_t^2 \|g_t\|^2 \right] \le \sum_{t=1}^{\infty} Q^2 t^{1-\eta} t^{-2+2\theta} = \sum_{t=1}^{\infty} O \left( \frac{1}{t^{1-2\theta+\eta}} \right) < \infty,$$

where $0 < \theta < \frac{1}{2}$ and $2\theta < \eta < 1$. This implies that the stepsize satisfying $\sum_{t=1}^{\infty} \eta_t^2 < \infty$ for SGD also satisfy the *Generalized Square Summability*, but the opposite direction might not be true. The proof of Proposition 3.1 is provided in Appendix C.1.

### 3.2 CONVERGENCE RATES FOR (FLEXADAGRAD-NORM) WITH STEPSIZES (TYPE I)

The following "descent inequality" in Lemma 3.2 is the foundation for obtaining the almost sure convergence rates with (Type I). The proof of Lemma 3.2 can be found in Appendix C.2.

**Lemma 3.2** (descent inequality for (FlexAdaGrad-Norm) with stepsize (Type I)). *Suppose Assumption 2.1 holds. Let $b > 0$, $0 < \epsilon \le \frac{1}{2}$ and $0 < \gamma \le 2$. Then for (FlexAdaGrad-Norm) with stepsizes (Type I), we have*

$$\mathbb{E}_t \left[ f(x_{t+1}) \right] \le f(x_t) - \mathbb{E}_t \left[ \eta_t \|\nabla f(x_t)\|^2 \right] + \frac{M}{2} \mathbb{E}_t \left[ \eta_t^2 \|g_t\|^2 \right]. \tag{2}$$

With this inequality, we provide almost sure convergence rates for (FlexAdaGrad-Norm) with (Type I) in strongly convex, convex and nonconvex cases, as shown in Theorem 3.3, 3.4 and 3.5 respectively. Firstly, we give the results of strongly convex case in Theorem 3.3, whose proof is in Appendix C.2.

**Theorem 3.3** (last-iterate convergence rate). *Suppose Assumptions 2.1 and 2.2 hold and $f(x)$ is $\mu$-strongly convex. Set the stepsizes as (Type I) for (FlexAdaGrad-Norm). Let $b > 0$, $0 < \epsilon \le \frac{1}{2}$, $0 < \eta < 1$ and $0 < \gamma < 2$. Set $\epsilon, \eta$ and $\gamma$ such that*

$$2(1 - \eta) \le 1 - \frac{\gamma}{2}, \quad b \ge \frac{Q^\gamma}{2^{\frac{1}{1+2\epsilon}} - 1}.$$

*Then almost surely,*

$$f(x_t) - f^* = o \left( \frac{1}{t^{1-\eta}} \right).$$

Next, we provide the almost sure convergence rate for (FlexAdaGrad-Norm) with (Type I) in the convex case. We highlight that the almost sure convergence in Theorem 3.4 is in the sense of last-iterate. Actually, even for convergence rates in expectation, there are no last-iterate convergence rates for the convex case in the existing literature. The existing literature for AdaGrad in the convex setting can only provide the average-iterate convergence rates (refers to Li & Orabona (2019); Levy (2017); Duchi et al. (2011); McMahan & Streeter (2010); Attia & Koren (2023)). Hence, we are the first to provide the **last-iterate** convergence rate in Theorem 3.4, whose proof is in Appendix C.2.

**Theorem 3.4** (last-iterate convergence rate). *Suppose Assumptions 2.1 and 2.2 hold and $f(x)$ is convex. Set the stepsizes as (Type I) for (FlexAdaGrad-Norm). Let $b > 0$, $0 < \epsilon < \frac{1}{2}$ and $0 < \gamma < 2$. Set $\epsilon$ and $\gamma$ such that*

$$\frac{1}{2} - \epsilon \leq \frac{1}{2}(1 - \frac{\gamma}{2}), \quad b \geq \frac{Q^\gamma}{2^{\frac{1}{1+2\epsilon}} - 1}.$$

*Then almost surely, there exists $\tilde{x} \in \arg\min_{x \in \mathbb{R}^d} f(x)$ such that*

$$\lim_{t \to \infty} x_t = \tilde{x} \quad and \quad f(x_t) - f^* = O\left(\frac{1}{t^{\frac{1}{2} - \epsilon}}\right).$$

Moving beyond convexity, we provide the almost sure convergence rate in the nonconvex case, as shown in Theorem 3.5. The proof of Theorem 3.5 is deferred to Appendix C.2.

**Theorem 3.5** (best-iterate convergence rate). *Suppose Assumptions 2.1 and 2.2 hold. Set the stepsizes as (Type I) for (FlexAdaGrad-Norm). Let $0 < \epsilon < \frac{1}{2}$, $0 < \gamma \leq 2$ and $b \geq \frac{Q^\gamma}{2^{\frac{1}{1+2\epsilon}} - 1}$. Then almost surely,*

$$\min_{1 \leq i \leq t} \|\nabla f(x_i)\|^2 = O\left(\frac{1}{t^{\frac{1}{2} - \epsilon}}\right).$$

## 3.3 CONVERGENCE RATES FOR (FLEXADAGRAD-NORM) WITH STEPSIZES (TYPE II)

The main difference between (Type I) and (Type II) lies in the descent inequality. The analysis of (Type II) relies on the descent inequality in Lemma 3.6, which is different from the descent inequality used for (Type I). The proofs of Lemma 3.6 can be found in Appendix C.3.

**Lemma 3.6** (descent inequality for (FlexAdaGrad-Norm) with stepsize (Type II)). *Suppose Assumptions 2.1 and 2.2 hold. Let $b > 0$, $0 < \epsilon \leq \frac{1}{2}$ and $0 < \gamma \leq 2$. Set $b \geq \max\left\{\frac{Q^\gamma}{2^{\frac{1}{1+2\epsilon}} - 1}, Q^2\right\}$. Then for (FlexAdaGrad-Norm) with stepsize (Type II), we have*

$$\mathbb{E}_t[f(x_{t+1})] \leq f(x_t) - \left[1 - \left(\frac{1}{2}\right)^{\frac{1+\epsilon}{2(1+2\epsilon)}}\right]\mathbb{E}_t\left[\eta_{t-1}\|\nabla f(x_t)\|^2\right]$$

$$+ \frac{M}{2}\mathbb{E}_t\left[\frac{\|g_t\|^2}{\left(b + \sum_{i=1}^t \|g_i\|^\gamma\right)^{1+2\epsilon}}\right] + \left(\frac{1}{2}\right)^{\frac{1+\epsilon}{2(1+2\epsilon)}}\mathbb{E}_t\left[\frac{Q^2\|g_t\|^2}{\left(b + \sum_{i=1}^t \|g_i\|^\gamma\right)^{1+\epsilon}}\right]. \tag{3}$$

With this lemma in hand, we can provide the almost sure convergence rates for (FlexAdaGrad-Norm) with the stepsize (Type II).

Firstly, we will provide **last-iterate** almost sure convergence rates in (strongly) convex cases, as shown in Theorem 3.7 and 3.8. The proofs of these theorems are deferred to Appendix C.3.

**Theorem 3.7** (last-iterate convergence). *Suppose Assumptions 2.1 and 2.2 hold and $f(x)$ is $\mu$-strongly convex. Set the stepsizes as (Type II) for (FlexAdaGrad-Norm). Let $b > 0$, $0 < \epsilon \leq \frac{1}{2}$, $0 < \eta < 1$ and $0 < \gamma < 2$. Set $b$, $\eta$ and $\gamma$ such that*

$$2(1 - \eta) \leq 1 - \frac{\gamma}{2}, \quad b \geq \max\left\{\frac{Q^\gamma}{2^{\frac{1}{1+2\epsilon}} - 1}, Q^2\right\}.$$

*Then almost surely,*

$$f(x_t) - f^* = o\left(\frac{1}{t^{1-\eta}}\right).$$

**Theorem 3.8** (last-iterate convergence). *Suppose Assumptions 2.1 and 2.2 hold and $f(x)$ is convex. Set the stepsizes as (Type II) for (FlexAdaGrad-Norm). Let $b > 0$, $0 < \epsilon < \frac{1}{2}$ and $0 < \gamma < 2$. Set $b$, $\epsilon$ and $\gamma$ such that*

$$\frac{1}{2} - \epsilon \leq \frac{1}{2}(1 - \frac{\gamma}{2}), \quad b \geq \max\left\{\frac{Q^\gamma}{2^{\frac{1}{1+2\epsilon}} - 1}, Q^2\right\}.$$

*Then almost surely,*

$$f(x_t) - f^* = O\left(\frac{1}{t^{\frac{1}{2} - \epsilon}}\right).$$

Moving beyond convexity, we provide the best-iterate almost sure convergence rates of (FlexAdaGrad-Norm) with (Type II) in 3.9, whose proof can be founded in Appendix C.3.

**Theorem 3.9** (best-iterate convergence). *Suppose Assumptions 2.1 and 2.2 hold. Set the stepsizes as (Type II) for (FlexAdaGrad-Norm). Let $0 < \epsilon < \frac{1}{2}$, $0 < \gamma \leq 2$ and $b \geq \max\left\{\frac{Q^\gamma}{2^{\frac{1}{1+2\epsilon}} - 1}, Q^2\right\}$. Then almost surely,*

$$\min_{1 \leq i \leq t} \|\nabla f(x_i)\|^2 = O\left(\frac{1}{t^{\frac{1}{2} - \epsilon}}\right).$$

**Remark 3.10.** *It is worth noting that the almost sure convergence rates obtained in Theorem 3.4 and 3.8 (Theorem 3.5 and 3.9) are almost close to the SOTA convergence rates in expectation of AdaGrad-Norm for convex (nonconvex) cases, up to an $\epsilon$-factor.*

It should be pointed out that we can not obtain the last-iterate convergence rates for the nonconvex case. However, the last-iterate convergence is of great theoretical and practical interest in many fields Golowich et al. (2020); Daskalakis & Panageas (2019). For the nonconvex case, Li & Orabona (2019) provided the asymptotically last-iterate almost sure convergence for AdaGrad with (Type I). We are not aware of any almost sure last-iterate convergence guarantee of (FlexAdaGrad-Norm) with (Type II) for the nonconvex case in existing work. To bridge this gap, we obtain the almost sure last-iterate convergence of (FlexAdaGrad-Norm) with (Type II) for nonconvex $f(x)$ in Proposition 3.11, whose proof can be found in Appendix C.3.

**Proposition 3.11** (last-iterate asymptotical convergence for non-convex case). *Suppose Assumptions 2.1 and 2.2 hold. Set stepsizes as (Type II) for (FlexAdaGrad-Norm). Let $b \geq \max\left\{\frac{Q^\gamma}{2^{\frac{1}{1+2\epsilon}} - 1}, Q^2\right\}$, where $\epsilon \in (0, \frac{1}{2}]$ and $\gamma \in (0, 2]$. Then almost surely,*

$$\lim_{t \to \infty} \|\nabla f(x_t)\| = 0.$$

## 4 CONCLUSION AND FUTURE DIRECTION

**Conclusion.** In this paper, we introduce a flexibility parameter $\gamma$ to control the impact of gradient history in original (AdaGrad-Norm) and propose an new variant (FlexAdaGrad-Norm). This parameter plays a central role in establishing almost sure convergence rates for AdaGrad. Specifically, we close the gap by providing the **last-iterate** convergence rates of (FlexAdaGrad-Norm) in the (strongly) convex cases. These convergence rates hold in the almost sure sense and at any time $t \geq 1$, thereby offering a stable and **robust** guarantee for (FlexAdaGrad-Norm). We also present the almost sure convergence rates of nonconvex cases in any time $t \geq 1$. Our analysis covers all stepsizes including (Type I) and (Type II) for AdaGrad in existing literature. Besides, we present an experiment on the real-world dataset to show the practical efficiency of $\gamma$. Hence, we provide some new insights for AdaGrad from both theoretical and practical points.

**Future direction.** Liu & Yuan (2022) provide a unified framework to analyze almost sure convergence rates of SGD and its two variants SHB and SNAG, whereas we only investigate AdaGrad in this paper. An important and interesting direction is to investigate the almost sure convergence rates for other algorithms with adaptive stepsizes such as Adam (Kingma & Ba, 2014) and AdamW (Loshchilov & Hutter, 2019). Another direction can be investigated in the future is obtaining almost sure convergence rates of AdaGrad (or its variants) with DIAG matrix adaption and FULL matrix adaption proposed in Duchi et al. (2011).

**Reproducibility Statement.** All theoretical results in this paper are accompanied by detailed mathematical proofs, and the assumptions are explicitly stated. Also, the code of experiment is provided in the supplementary material.

**The Use of Large Language Models.** A large language model was utilized for grammar checking and polishing during its writing process in this paper. The authors have checked all content generated by the large language model and confirm and take responsibility for it.

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

## A    MORE DETAILS OF EXPERIMENTS

We conducted experiments using the VGG+BN+Dropout network in Wilson et al. (2017) on the CIFAR-10 dataset implemented in PyTorch. The experiments were run with a fixed budget of 150 epochs using an NVIDIA GeForce RTX 3090 GPU and an Intel Core i9-10900K CPU @ 3.70GHz.

Figure 1 shows the learning history of AdaGrad with varying $\gamma$ and $b$ values on the training and testing datasets. Overall, after introducing the flexibility parameter $\gamma$, AdaGrad possesses improved performance as $\gamma$ decreases. As for the configuration $\gamma = 2$, which consistently worse relative to other settings ($\gamma = 0.01, 0.1$ and $1$), resulting in higher loss and lower accuracy both on the training and testing datasets. Additionally, the curve convergence process for the configuration $\gamma = 2$ is notably unstable in both the Train and Test Accuracy plots compared to other settings ($\gamma = 0.01, 0.1$ and $1$). This instability is particularly evident in the Test Loss figure (bottom right of Figure 1), which exhibits sharp peaks and troughs during the first 50 epochs. According to the bottom right of Figure 1, the curve fluctuates more sharply for larger $\gamma$ values, resulting in increased instability in the convergence process for the configuration $\gamma = 2$. These findings clearly demonstrate the practical efficiency of $0 < \gamma < 2$, reinforcing our motivation that it can provide more flexibility for practitioners when they apply AdaGrad to real-world problems.

## B    TECHNICAL LEMMAS

**Lemma B.1** (Theorem 2.1.10 in Nesterov et al. (2018)). *If $f(x)$ is $\mu$-strongly convex, then*

$$\frac{1}{2\mu}\|\nabla f(x)\|^2 \geq f(x) - f^*.$$

**Lemma B.2** (Lemma 2 in Li & Orabona (2019)). *Let $r_0 > 0$, $r_n \geq 0$ and $e > 1$. Then*

$$\sum_{t=1}^{\infty} \frac{r_t}{\left(r_0 + \sum_{i=1}^{t} r_i\right)^e} \leq \frac{1}{(e-1)r_0^{e-1}}.$$

**Lemma B.3** (Proposition 2 in Alber et al. (1998)). *Let $\{r_t\}$, $\{s_t\}$ be two non-negative sequences. Assume that $\sum_{t=1}^{\infty} r_n s_n$ converges and $\sum_{t=1}^{\infty} r_t$ diverges, i.e.,*

$$\sum_{n=1}^{\infty} r_t s_t < \infty \ \text{ and } \ \sum_{t=1}^{\infty} r_n = \infty.$$

*Besides, if there exists $K \geq 0$ such that*

$$\|s_{t+1} - s_t\| \leq K r_t.$$

*Then it holds that*

$$\lim_{t \to \infty} s_n = 0.$$

**Lemma B.4.** *For any $0 < q \leq 1$ and any positive integer $t$, we have*

$$(t+1)^q \leq t^q + q\, t^{q-1}$$

*Proof.* Applying the mean value theorem for the function $h(x) = x^q$, we have

$$(t+1)^q - t^q = q\varrho^{q-1} \leq qt^{q-1}$$

where $\varrho \in (t, t+1)$. We complete the proof. $\square$

**Lemma B.5** (Borel-Cantelli Lemma). *If the sum of the probabilities of the events $\{E_n\}$ is finite, i.e.,*

$$\sum_{i=1}^{\infty} \mathbb{P}(E_n) < \infty,$$

*then the probability that infinite many of them occur is 0, i.e.,*

$$\mathbb{P}\left(\limsup_{n \to \infty} E_n\right) = 0.$$

## C    PROOF OF SECTION 3

In this section, we present the detailed proofs of Section 3.

### C.1    PROOF OF SECTION 3.1

*Proof of Proposition 3.1.* Let $\mathbb{N}^+$ be the set of positive integers. Define

$$\mathcal{A}_1 = \{i \in \mathbb{N}^+ \mid \|g_i\|^2 \geq \frac{1}{\sqrt{i}}\} \quad \text{and} \quad \mathcal{A}_2 = \{i \in \mathbb{N}^+ \mid \|g_i\|^2 < \frac{1}{\sqrt{i}}\}.$$

There are two cases for proving $\sum_{t=1}^{\infty} \mathbb{E}_t \left[ t^{1-\eta} \frac{\|g_t\|^2}{\left(b + \sum_{i=1}^{t} \|g_i\|^{\gamma}\right)^{1+\beta}} \right] < \infty$ in the following:

(I)  all $t \in \mathcal{A}_2$, i.e., $\mathcal{A}_2 = \mathbb{N}^+$;

(II)  not all $t \in \mathcal{A}_2$, i.e., $\mathbb{N}^+ = \mathcal{A}_1 \cup \mathcal{A}_2$ and $\mathcal{A}_1 \neq \emptyset$.

The proof of $\sum_{t=1}^{\infty} \mathbb{E}_t \left[ t^{1-\eta} \frac{\|g_t\|^2}{\left(b + \sum_{i=1}^{t} \|g_i\|^{\gamma}\right)^{1+\beta}} \right] < \infty$ can be decomposed as the following steps:

- Step 1: Select appropriate parameters $\epsilon$, $\eta$ and $\gamma$ to prove for the extreme Case (I);

- Step 2: The main difficult of Case (II) is that there exists $t \in \mathcal{A}_1$ such that some terms $t^{1-\eta} \frac{\|g_t\|^2}{\left(b + \sum_{i=1}^{t} \|g_i\|^{\gamma}\right)^{1+\beta}}$ too large, we can not directly bound these terms. To prove Case (II), for $t \in \mathcal{A}_1$, we first decompose $\|g_t\|^{\gamma}$ into the sum of some terms $\{\|\tilde{g}_j\|^{\gamma}\}_j$ such that all $j \in \mathcal{A}_2$. With this decomposition in hand, we prove that $\sum_{t=1}^{\infty} \mathbb{E}_t \left[ t^{1-\eta} \frac{\|g_t\|^2}{\left(b + \sum_{i=1}^{t} \|g_i\|^{\gamma}\right)^{1+\beta}} \right]$ in Case (II) can be bounded by the result in Case (I).

For the convenience of proof, we define the following notation:

- If $t \in \mathcal{A}_1$, denote $g_t$ by $\bar{g}_t$; If $t \in \mathcal{A}_2$, denote $g_t$ by $\hat{g}_t$.

**Proof of Case (I)**: If all $t \in \mathcal{A}_2$, using the fact that one can exchange infinite sum and expectation if the terms are nonnegative, we have

$$\sum_{t=1}^{\infty} \mathbb{E}_t \left[ t^{1-\eta} \frac{\|\hat{g}_t\|^2}{\left(b + \sum_{i=1}^{t} \|\hat{g}_i\|^{\gamma}\right)^{1+\beta}} \right]$$

$$= \mathbb{E}_t \left[ \sum_{t=1}^{\infty} t^{1-\eta} \frac{\|\hat{g}_t\|^2}{\left(b + \sum_{i=1}^{t} \|\hat{g}_i\|^{\gamma}\right)^{1+\beta}} \right]$$

$$\leq \mathbb{E}_t \left[ \sum_{t=1}^{\infty} \|\hat{g}_t\|^{-4(1-\eta)} \frac{\|\hat{g}_t\|^2}{\left(b + \sum_{i=1}^{t} \|\hat{g}_i\|^{\gamma}\right)^{1+\beta}} \right] \tag{4}$$

$$= \mathbb{E}_t \left[ \sum_{t=1}^{\infty} \|\hat{g}_t\|^{2-4(1-\eta)-\gamma} \frac{\|\hat{g}_t\|^{\gamma}}{\left(b + \sum_{i=1}^{t} \|\hat{g}_i\|^{\gamma}\right)^{1+\beta}} \right]$$

$$\leq \mathbb{E}_t \left[ \sum_{t=1}^{\infty} \frac{\|\hat{g}_t\|^{\gamma}}{\left(b + \sum_{i=1}^{t} \|\hat{g}_i\|^{\gamma}\right)^{1+\beta}} \right]$$

$$< \infty,$$

where we use the fact $2(1-\eta) \le 1 - \frac{\gamma}{2}$ and $\|\hat{g}_t\| < \frac{1}{\sqrt{t^{\frac{1}{2}}}} \le 1$ in the second inequality; we use Lemma B.2 in the third inequality.

**Proof of Case (II):** Note that in the Case (II), we can get

$$
\sum_{t=1}^{\infty} \mathbb{E}_t \left[ t^{1-\eta} \frac{\|g_t\|^2}{\left(b + \sum_{i=1}^{t} \|g_i\|^\gamma\right)^{1+\beta}} \right]
$$

$$
= \mathbb{E}_t \left[ \sum_{t=1}^{\infty} t^{1-\eta} \frac{\|g_t\|^2}{\left(b + \sum_{i=1}^{t} \|g_i\|^\gamma\right)^{1+\beta}} \right]
$$

$$
= \mathbb{E}_t \left[ \sum_{t \in \mathcal{A}_1} t^{1-\eta} \frac{\|g_t\|^2}{\left(b + \sum_{i=1}^{t} \|g_i\|^\gamma\right)^{1+\beta}} \right] + \mathbb{E}_t \left[ \sum_{t \in \mathcal{A}_2} t^{1-\eta} \frac{\|g_t\|^2}{\left(b + \sum_{i=1}^{t} \|g_i\|^\gamma\right)^{1+\beta}} \right] \quad (5)
$$

$$
= \mathbb{E}_t \left[ \sum_{t \in \mathcal{A}_1} t^{1-\eta} \frac{\|g_t\|^\gamma \|g_t\|^{2-\gamma}}{\left(b + \sum_{i=1}^{t} \|g_i\|^\gamma\right)^{1+\beta}} \right] + \mathbb{E}_t \left[ \sum_{t \in \mathcal{A}_2} t^{1-\eta} \frac{\|g_t\|^\gamma \|g_t\|^{2-\gamma}}{\left(b + \sum_{i=1}^{t} \|g_i\|^\gamma\right)^{1+\beta}} \right]
$$

$$
\le \mathbb{E}_t \left[ \sum_{t \in \mathcal{A}_1} \frac{D_3\, t^{1-\eta}\, \|\bar{g}_t\|^\gamma}{\left(b + \sum_{i=1}^{t} \|g_i\|^\gamma\right)^{1+\beta}} \right] + \mathbb{E}_t \left[ \sum_{t \in \mathcal{A}_2} \frac{D_3\, t^{1-\eta}\, \|\hat{g}_t\|^\gamma}{\left(b + \sum_{i=1}^{t} \|g_i\|^\gamma\right)^{1+\beta}} \right],
$$

where we can exchange infinite sum and expectation since the terms are nonnegative in the first equality and let $D_3 = Q^{2-\gamma}$ in the inequality.

We use the decomposition skill as stated below.

(i) If $t \in \mathcal{A}_1$, we can always decompose $\|g_t\|^\gamma$ into the sum of $k_t$ terms $\{\|\tilde{g}_j\|^\gamma\}_j$ such that all $j \in \mathcal{A}_2$. Concretely, for $\|g_t\|^\gamma$, there exists $k_t$ terms such that

$$
\|g_t\|^\gamma = \sum_{j=S_{t-1}+1}^{S_t} \|\tilde{g}_j\|^\gamma, \quad S_t = \sum_{i=1}^{t} k_i, \quad (6)
$$

where $S_0 = 0$ and $\|\tilde{g}_j\|^2 < \frac{1}{\sqrt{j}}$ for $j = S_{t-1}+1, S_{t-1}+2, \cdots, S_t$.

(ii) If $t \in \mathcal{A}_2$, set $k_t = 1$ and we do not need to decompose $\|g_t\|^\gamma$.

After the above decomposition, for any $t \ge 1$, we have decomposed $\{\|g_i\|^\gamma\}_{i=1}^{t}$ into $\{\|\tilde{g}_j\|^\gamma\}_{j=1}^{S_t}$ such that $j \in \mathcal{A}_2$ for $j = 1, 2, \cdots, S_t$ and

$$
\sum_{i=1}^{t} \|g_i\|^\gamma = \sum_{j=1}^{S_t} \|\tilde{g}_j\|^\gamma. \quad (7)
$$

Using (6) and (7), we have

$$
\frac{\|g_t\|^\gamma}{\left(b + \sum_{i=1}^{t} \|g_i\|^\gamma\right)^{1+\beta}} = \sum_{j=S_{t-1}+1}^{S_t} \frac{\|\tilde{g}_j\|^\gamma}{\left(b + \sum_{i=1}^{S_t} \|\tilde{g}_i\|^\gamma\right)^{1+\beta}}
$$

$$
\le \sum_{j=S_{t-1}+1}^{S_t} \frac{\|\tilde{g}_j\|^\gamma}{\left(b + \sum_{i=1}^{j} \|\tilde{g}_i\|^\gamma\right)^{1+\beta}}, \quad (8)
$$

where in the inequality, we use the fact that for $j = S_{t-1}+1, S_{t-1}+2, \cdots, S_t$, it satisfies

$$
\frac{1}{\left(b + \sum_{i=1}^{S_t} \|\tilde{g}_i\|^\gamma\right)^{1+\beta}} \le \frac{1}{\left(b + \sum_{i=1}^{j} \|\tilde{g}_i\|^\gamma\right)^{1+\beta}}.
$$

Also note that $j^{1-\eta} \geq t^{1-\eta}$ for $j = S_{t-1} + 1, S_{t-1} + 2, \cdots, S_t$.

Combining the above, for any $t$, we can obtain

$$\frac{t^{1-\eta} \|g_t\|^{\gamma}}{\left(b + \sum_{i=1}^{t} \|g_i\|^{\gamma}\right)^{1+\beta}} \leq \sum_{j=S_{t-1}+1}^{S_t} \frac{j^{1-\eta} \|\tilde{g}_j\|^{\gamma}}{\left(b + \sum_{i=1}^{j} \|\tilde{g}_i\|^{\gamma}\right)^{1+\beta}}. \tag{9}$$

Then for any $m \in \mathbb{N}^+$, we have

$$\sum_{t=1}^{m} \frac{t^{1-\eta} \|g_t\|^{\gamma}}{\left(b + \sum_{i=1}^{t} \|g_i\|^{\gamma}\right)^{1+\beta}} \leq \sum_{t=1}^{m} \sum_{j=S_{t-1}+1}^{S_t} \frac{j^{1-\eta} \|\tilde{g}_j\|^{\gamma}}{\left(b + \sum_{i=1}^{j} \|\tilde{g}_i\|^{\gamma}\right)^{1+\beta}}$$

$$= \sum_{j=1}^{S_m} \frac{j^{1-\eta} \|\tilde{g}_j\|^{\gamma}}{\left(b + \sum_{i=1}^{j} \|\tilde{g}_i\|^{\gamma}\right)^{1+\beta}}. \tag{10}$$

Taking the limit for $m \to \infty$, we have

$$\sum_{t=1}^{\infty} \frac{t^{1-\eta} \|g_t\|^{\gamma}}{\left(b + \sum_{i=1}^{t} \|g_i\|^{\gamma}\right)^{1+\beta}} \leq \sum_{j=1}^{\infty} \frac{j^{1-\eta} \|\tilde{g}_j\|^{\gamma}}{\left(b + \sum_{i=1}^{j} \|\tilde{g}_i\|^{\gamma}\right)^{1+\beta}}. \tag{11}$$

Note that from the above stated decomposition, $\|\tilde{g}_j\|^2 < \frac{1}{\sqrt{j}}$ for all $j$, then we can get $\sum_{j=1}^{\infty} \frac{j^{1-\eta} \|\tilde{g}_j\|^{\gamma}}{\left(b + \sum_{i=1}^{j} \|\tilde{g}_i\|^{\gamma}\right)^{1+\beta}} < \infty$ from the result of Case (I). Then we complete all the proofs. $\qquad\square$

## C.2    PROOF OF SECTION 3.2

*Proof of Lemma 3.2.* As $f(x)$ is $M$-smooth, we can get

$$f(x_{t+1}) \leq f(x_t) + \langle \nabla f(x_t), x_{t+1} - x_t \rangle + \frac{M}{2} \|x_{t+1} - x_t\|^2$$

$$= f(x_t) + \langle \nabla f(x_t), -\eta_t g_t \rangle + \frac{M}{2} \eta_t^2 \|g_t\|^2. \tag{12}$$

Taking the conditional expectation with respect to $\xi_1, \xi_2, ..., \xi_t$, we can obtain

$$\mathbb{E}_t [f(x_{t+1})] \leq f(x_t) - \mathbb{E}_t \left[ \eta_t \|\nabla f(x_t)\|^2 \right] + \frac{M}{2} \mathbb{E}_t \left[ \eta_t^2 \|g_t\|^2 \right]. \tag{13}$$

This completes the proof. $\qquad\square$

*Proof of Theorem 3.3.* Note that by Assumption 2.2, we have $\|g_i\| \leq Q$, then

$$\eta_t = \frac{1}{\left(b + \sum_{i=1}^{t-1} \|g_i\|^{\gamma}\right)^{\frac{1}{2}+\epsilon}}$$

$$\geq \frac{1}{\left(b + \sum_{i=1}^{t-1} Q^{\gamma}\right)^{\frac{1}{2}+\epsilon}} \geq K_1 \frac{1}{t^{\frac{1}{2}+\epsilon}}, \tag{14}$$

where $K_1$ is a constant, i.e., there exists $K_1 > 0$ such that $\eta_t \geq K_1 \frac{1}{t^{\frac{1}{2}+\epsilon}}$.

Substituting this into (2), we can get

$$\mathbb{E}_t [f(x_{t+1})] \leq f(x_t) - \frac{K_1}{t^{\frac{1}{2}+\epsilon}} \|\nabla f(x_t)\|^2 + \frac{M}{2} \mathbb{E}_t \left[ \eta_t^2 \|g_t\|^2 \right]$$

$$\leq f(x_t) - \frac{K_1}{t^{\frac{1}{2}+\epsilon}} 2\mu \left( f(x_t) - f^* \right) + \frac{M}{2} \mathbb{E}_t \left[ \eta_t^2 \|g_t\|^2 \right], \tag{15}$$

where we used Lemma B.1 in the second inequality. Then we can get

$$\mathbb{E}_t\left[f(x_{t+1}) - f^*\right] \le \left(1 - \frac{2\mu K_1}{t^{\frac{1}{2}+\epsilon}}\right)(f(x_t) - f^*) + \frac{M}{2}\mathbb{E}_t\left[\eta_t^2\|g_t\|^2\right]. \tag{16}$$

Multiplying by $(t+1)^{1-\eta}$ in two sides of (16), we can get

$$\mathbb{E}_t\left[(t+1)^{1-\eta}(f(x_{t+1}) - f^*)\right]$$

$$\le \left(1 - \frac{2\mu K_1}{t^{\frac{1}{2}+\epsilon}}\right)(t+1)^{1-\eta}(f(x_t) - f^*) + \frac{M}{2}\mathbb{E}_t\left[(t+1)^{1-\eta}\ \eta_t^2\|g_t\|^2\right]$$

$$\le \left(1 - \frac{2\mu K_1}{t^{\frac{1}{2}+\epsilon}}\right)\left[t^{1-\eta} + (1-\eta)t^{-\eta}\right](f(x_t) - f^*) + \frac{M}{2}\mathbb{E}_t\left[(t+1)^{1-\eta}\ \eta_t^2\|g_t\|^2\right] \tag{17}$$

$$= \left(1 - \frac{2\mu K_1}{t^{\frac{1}{2}+\epsilon}}\right)\left[1 + \frac{1-\eta}{t}\right]t^{1-\eta}(f(x_t) - f^*) + \frac{M}{2}\mathbb{E}_t\left[(t+1)^{1-\eta}\ \eta_t^2\|g_t\|^2\right]$$

$$\le t^{1-\eta}(f(x_t) - f^*) - \frac{D}{t^{\frac{1}{2}+\epsilon}}t^{1-\eta}(f(x_t) - f^*) + \frac{M}{2}\mathbb{E}_t\left[(t+1)^{1-\eta}\ \eta_t^2\|g_t\|^2\right],$$

where we use Lemma B.4 in the second inequality; in the third inequality, $D > 0$ is a constant and we use the fact that the domination term of $\frac{1-\eta}{t} - \frac{2\mu K_1(1-\eta)}{t^{\frac{3}{2}+\epsilon}} - \frac{2\mu K_1}{t^{\frac{1}{2}+\epsilon}}$ is $-\frac{2\mu K_1}{t^{\frac{1}{2}+\epsilon}}$.

Recall that for all $i \in \mathbb{N}^+$, it satisfies $\|g_i\| \le Q$. Then we have

$$\frac{\eta_t^2}{\eta_{t+1}^2} = \frac{\left(b + \sum_{i=1}^t\|g_i\|^\gamma\right)^{1+2\epsilon}}{\left(b + \sum_{i=1}^{t-1}\|g_i\|^\gamma\right)^{1+2\epsilon}}$$

$$= \left(1 + \frac{\|g_t\|^\gamma}{b + \sum_{i=1}^{t-1}\|g_i\|^\gamma}\right)^{1+2\epsilon} \tag{18}$$

$$\le \left(1 + \frac{Q^\gamma}{b}\right)^{1+2\epsilon} \le 2.$$

With this inequality in hand, we have

$$\mathbb{E}_t\left[(t+1)^{1-\eta}\ \eta_t^2\|g_t\|^2\right] \le 2\,\mathbb{E}_t\left[(t+1)^{1-\eta}\ \eta_{t+1}^2\|g_t\|^2\right]$$

$$\le 2\,K_2\,\mathbb{E}_t\left[t^{1-\eta}\ \eta_{t+1}^2\|g_t\|^2\right]$$

$$= 2\,K_2\,\mathbb{E}_t\left[t^{1-\eta}\frac{\|g_t\|^2}{\left(b + \sum_{i=1}^t\|g_i\|^\gamma\right)^{1+2\epsilon}}\right], \tag{19}$$

where we can select $K_2 > 0$ (e.g. $K_2 = 2$) such that $(t+1)^{1-\eta} \le K_2 t^{1-\eta}$ in the second inequality.

Putting (17) and (19) together, we have

$$\mathbb{E}_t\left[(t+1)^{1-\eta}(f(x_{t+1}) - f^*)\right]$$

$$\le t^{1-\eta}(f(x_t) - f^*) - \frac{Dt^{1-\eta}}{t^{\frac{1}{2}+\epsilon}}(f(x_t) - f^*) + MK_2\,\mathbb{E}_t\left[\frac{t^{1-\eta}\|g_t\|^2}{\left(b + \sum_{i=1}^t\|g_i\|^\gamma\right)^{1+2\epsilon}}\right]. \tag{20}$$

By Proposition 3.1 and (19), we can immediately get

$$\sum_{t=1}^\infty \mathbb{E}_t\left[\frac{t^{1-\eta}\|g_t\|^2}{\left(b + \sum_{i=1}^t\|g_i\|^\gamma\right)^{1+2\epsilon}}\right] < \infty$$

and

$$\sum_{t=1}^{\infty} \mathbb{E}_t \left[ (t+1)^{1-\eta} \, \eta_t^2 \|g_t\|^2 \right] < \infty.$$

Applying Lemma 2.4 for (20) with

$$Y_t = t^{1-\eta} \left( f(x_t) - f^* \right), \quad X_t = \frac{Dt^{1-\eta}}{t^{\frac{1}{2}+\epsilon}} \left( f(x_t) - f^* \right),$$

and

$$Z_t = MK_2 \, \mathbb{E}_t \left[ \frac{t^{1-\eta} \|g_t\|^2}{\left( b + \sum_{i=1}^{t} \|g_i\|^{\gamma} \right)^{1+2\epsilon}} \right], \quad \theta_t = 0,$$

we can immediately get the sequence $\{ t^{1-\eta} \left( f(x_t) - f^* \right) \}$ converges almost surely and

$$\sum_{t=1}^{\infty} \frac{1}{t^{\frac{1}{2}+\epsilon}} t^{1-\eta} \left( f(x_t) - f^* \right) < \infty$$

almost surely. Together with $\sum_{t=1}^{\infty} \frac{1}{t^{\frac{1}{2}+\epsilon}} = \infty$, we can get

$$\lim_{t \to \infty} t^{1-\eta} \left( f(x_t) - f^* \right) = 0$$

almost surely. We complete the proof of Theorem 3.3. $\qquad \square$

*Proof of Theorem 3.4.* Multiplying by $(t+1)^{\frac{1}{2}-\epsilon}$ in two sides of (2) and using Lemma B.4, we can get

$$\mathbb{E}_t \left[ (t+1)^{\frac{1}{2}-\epsilon} f(x_{t+1}) - f^* \right]$$

$$\leq (t+1)^{\frac{1}{2}-\epsilon} \left( f(x_t) - f^* \right) - \mathbb{E}_t \left[ (t+1)^{\frac{1}{2}-\epsilon} \eta_t \|\nabla f(x_t)\|^2 \right] + \frac{M}{2} \mathbb{E}_t \left[ (t+1)^{\frac{1}{2}-\epsilon} \eta_t^2 \|g_t\|^2 \right]$$

$$\leq \left[ t^{\frac{1}{2}-\epsilon} + \left( \frac{1}{2} - \epsilon \right) t^{-\frac{1}{2}-\epsilon} \right] \left( f(x_t) - f^* \right) + \frac{M}{2} \mathbb{E}_t \left[ (t+1)^{\frac{1}{2}-\epsilon} \eta_t^2 \|g_t\|^2 \right]$$

$$= t^{\frac{1}{2}-\epsilon} \left( f(x_t) - f^* \right) + \left( \frac{1}{2} - \epsilon \right) \frac{1}{t^{\frac{1}{2}+\epsilon}} \left( f(x_t) - f^* \right) + \frac{M}{2} \mathbb{E}_t \left[ (t+1)^{\frac{1}{2}-\epsilon} \eta_t^2 \|g_t\|^2 \right].$$

$$(21)$$

By direct computation, we have

$$\mathbb{E}_t \left[ \|x_{t+1} - x^*\|^2 \right] = \mathbb{E}_t \left[ \|x_{t+1} - x_t + x_t - x^*\|^2 \right]$$

$$= \|x_t - x^*\|^2 - 2\mathbb{E}_t \left[ \langle \eta_t g_t, x_t - x^* \rangle \right] + \mathbb{E}_t \left[ \eta_t^2 \|g_t\|^2 \right] \qquad (22)$$

$$= \|x_t - x^*\|^2 - 2 \langle \eta_t \nabla f(x_t), x_t - x^* \rangle + \mathbb{E}_t \left[ \eta_t^2 \|g_t\|^2 \right].$$

For any minimizer $x^*$, by the convexity of $f(x)$ and the proven fact $\eta_t \geq \frac{K_1}{t^{\frac{1}{2}+\epsilon}}$, we have

$$\mathbb{E}_t \left[ \|x_{t+1} - x^*\|^2 \right] \leq \|x_t - x^*\|^2 - 2\eta_t \left( f(x_t) - f^* \right) + \mathbb{E}_t \left[ \eta_t^2 \|g_t\|^2 \right]$$

$$\leq \|x_t - x^*\|^2 - 2\frac{K_1}{t^{\frac{1}{2}+\epsilon}} \left( f(x_t) - f^* \right) + \mathbb{E}_t \left[ \eta_t^2 \|g_t\|^2 \right]. \qquad (23)$$

From the proven fact $\frac{\eta_t^2}{\eta_{t+1}^2} \leq 2$ in (18) and Lemma B.2, we can obtain $\sum_{t=1}^{\infty} \mathbb{E}_t \left[ \eta_t^2 \|g_t\|^2 \right] < \infty$. Applying Lemma 2.4 for (23) with

$$Y_t = \|x_t - x^*\|^2, X_t = \frac{2K_1}{t^{\frac{1}{2}+\epsilon}} \left( f(x_t) - f^* \right), Z_t = \mathbb{E}_t \left[ \eta_t^2 \|g_t\|^2 \right] \text{ and } \theta_t = 0,$$

we have $\{ \|x_t - x^*\| \}$ converges almost surely and

$$\sum_{t=1}^{\infty} \frac{1}{t^{\frac{1}{2}+\epsilon}} \left( f(x_t) - f^* \right) < \infty$$

almost surely. Replacing $1 - \eta$ with $\frac{1}{2} - \epsilon$ in Proposition 3.1, we can immediately get (Recall $(t+1)^{1-\eta} \le K_2 t^{1-\eta}$)

$$\sum_{t=1}^{\infty} \mathbb{E}_t \left[ (t+1)^{\frac{1}{2}-\epsilon} \eta_t^2 \|g_t\|^2 \right] \le K_2 \sum_{t=1}^{\infty} \mathbb{E}_t \left[ t^{\frac{1}{2}-\epsilon} \eta_t^2 \|g_t\|^2 \right] < \infty.$$

Applying Lemma 2.4 for (21) with

$$Y_t = t^{\frac{1}{2}-\epsilon} \left( f(x_t) - f^* \right), \quad X_t = 0, \quad \theta_t = 0,$$

and

$$Z_t = \left( \frac{1}{2} - \epsilon \right) \frac{1}{t^{\frac{1}{2}+\epsilon}} \left( f(x_t) - f^* \right) + \frac{M}{2} \mathbb{E}_t \left[ (t+1)^{\frac{1}{2}-\epsilon} \eta_t^2 \|g_t\|^2 \right],$$

we have the sequence $\left\{ t^{\frac{1}{2}-\epsilon} \left( f(x_t) - f^* \right) \right\}$ converges almost surely, i.e.,

$$f(x_t) - f^* = O \left( \frac{1}{t^{\frac{1}{2}-\epsilon}} \right)$$

almost surely. This means $\lim_{t \to \infty} f(x_t) = f^*$ almost surely.

Then we will prove there exists $\tilde{x} \in \arg\min_{x \in \mathbb{R}^d} f(x)$ such that $\lim_{t \to \infty} x_t = \tilde{x}$. Since we have got $\{\|x_t - x^*\|\}$ converges almost surely (a.s.), then $\{\|x_t - x^*\|\}$ is a.s. bounded and hence $\{x_t\}$ is a.s. bounded. Then there exists a subsequence $\{x_{t_k}\}$ converging to a point a.s. (denote the limit point by $\tilde{x}$). From the continuity of $f(x)$ and the proven fact $\lim_{t \to \infty} f(x_t) = f^*$ a.s., we have

$$f^* = \lim_{k \to \infty} f(x_{t_k}) = f \left( \lim_{k \to \infty} x_{t_k} \right) = f(\tilde{x}), \tag{24}$$

i.e., $\tilde{x} \in \arg\min_{x \in \mathbb{R}^d} f(x)$, thus we can set $x^* = \tilde{x}$. Recall that $\{\|x_t - x^*\|\}$ converges almost surely, then we have $\{\|x_t - \tilde{x}\|\}$ converges almost surely. From (24) and the continuity of $f(x)$, we conclude that there exists a convergent subsequence $\{\|x_{t_k} - \tilde{x}\|\}$ of $\{\|x_t - \tilde{x}\|\}$ satisfying $\lim_{k \to \infty} \|x_{t_k} - \tilde{x}\| = 0$ almost surely, together with the proven fact that $\{\|x_t - \tilde{x}\|\}$ converges almost surely, we can get

$$\lim_{t \to \infty} \|x_t - \tilde{x}\| = 0$$

almost surely. We complete all the proofs. $\qquad\square$

*Proof of Theorem 3.5.* For $t \in \mathbb{N}$, we define

$$w_t = \frac{2\eta_t}{\sum_{j=0}^{t} \eta_j}, \quad h_0 = \|\nabla f(x_0)\|^2, \quad h_{t+1} = (1 - w_t)h_t + w_t \|\nabla f(x_t)\|^2. \tag{25}$$

From (25), we can get

$$\|\nabla f(x_t)\|^2 = \frac{h_{t+1}}{w_t} - \left( \frac{1}{w_t} - 1 \right) h_t. \tag{26}$$

Substituting (26) into (2), we have

$$\mathbb{E}_t \left[ f(x_{t+1}) - f^* \right] \le f(x_t) - f^* - \eta_t \left[ \frac{h_{t+1}}{w_t} - \left( \frac{1}{w_t} - 1 \right) h_t \right] + \frac{M}{2} \mathbb{E}_t \left[ \eta_t^2 \|g_t\|^2 \right]$$

$$\le f(x_t) - f^* - \frac{\sum_{j=0}^{t} \eta_j}{2} h_{t+1} + \frac{\sum_{j=0}^{t} \eta_j}{2} h_t - \eta_t h_t + \frac{M}{2} \mathbb{E}_t \left[ \eta_t^2 \|g_t\|^2 \right] \tag{27}$$

Reorganizing it, we have

$$\mathbb{E}_t \left[ f(x_{t+1}) - f^* \right] + \frac{\sum_{j=0}^{t} \eta_j}{2} h_{t+1} \le f(x_t) - f^* + \frac{\sum_{j=0}^{t-1} \eta_j}{2} h_t - \frac{\eta_t h_t}{2} + \frac{M}{2} \mathbb{E}_t \left[ \eta_t^2 \|g_t\|^2 \right]. \tag{28}$$

We have proven $\sum_{t=1}^{\infty} \mathbb{E}_t \left[ \eta_t^2 \|g_t\|^2 \right] < \infty$ in the previous. Applying Lemma 2.4 for (28) with

$$Y_t = f(x_t) - f^* + \frac{\sum_{j=0}^{t-1} \eta_j}{2} h_t, \ X_t = \frac{\eta_t h_t}{2}, \ Z_t = \frac{M}{2} \mathbb{E}_t \left[ \eta_t^2 \|g_t\|^2 \right] \ \text{and} \ \theta_t = 0,$$

we can get

$$\{f(x_t) - f^*\} \quad \text{and} \quad \left\{ \left( \sum_{j=0}^{t-1} \eta_j \right) h_t \right\}$$

converges almost surely. Together with $\sum_{t=1}^{\infty} \eta_t = \infty$, we can get

$$h_t = O\left( \frac{1}{\sum_{j=0}^{t-1} \eta_j} \right) = O\left( \frac{1}{t^{\frac{1}{2}-\epsilon}} \right)$$

almost surely. Note that $h_t$ is the weighted average of $\left\{ \|\nabla f(x_t)\|^2 \right\}$, then we have

$$\min_{1 \le i \le t} \|\nabla f(x_i)\|^2 \le h_t.$$

We complete the proof. □

### C.3 PROOF OF SECTION 3.3

*Proof of Lemma 3.6.* Motivated by Wang et al. (2023), we can use smoothness of $f(x)$ to get Ada-Grad with stepsize (Type II) as follows

$$\mathbb{E}_t[f(x_{t+1})]$$

$$\le f(x_t) + \mathbb{E}_t \left[ \langle \nabla f(x_t), x_{t+1} - x_t \rangle + \frac{M}{2} \|x_{t+1} - x_t\|^2 \right]$$

$$= f(x_t) + \underbrace{\mathbb{E}_t[\langle \nabla f(x_t), -\eta_t g_t \rangle]}_{\text{First Order}} + \underbrace{\frac{M}{2} \mathbb{E}_t[\eta_t^2 \|g_t\|^2]}_{\text{Second Order}} \tag{29}$$

$$= f(x_t) + \underbrace{\mathbb{E}_t[\langle \nabla f(x_t), -\eta_{t-1} g_t \rangle]}_{\text{First Order Main}} + \underbrace{\mathbb{E}_t[\langle \nabla f(x_t), (\eta_{t-1} - \eta_t) g_t \rangle]}_{\text{Error}} + \underbrace{\frac{M}{2} \mathbb{E}_t[\eta_t^2 \|g_t\|^2]}_{\text{Second Order}}$$

$$= f(x_t) + \underbrace{\left( -\eta_{t-1} \|\nabla f(x_t)\|^2 \right)}_{\text{First Order Main}} + \underbrace{\mathbb{E}_t[\langle \nabla f(x_t), (\eta_{t-1} - \eta_t) g_t \rangle]}_{\text{Error}} + \underbrace{\frac{M}{2} \mathbb{E}_t[\eta_t^2 \|g_t\|^2]}_{\text{Second Order}}.$$

Obviously, the terms "**First Order Main**" and "**Second Order Main**" can be solved easily. The main difficulty of analysis focuses on the "**Error**" term due to the coupling relation between $\eta_t$ and $g_t$. Next, we will make a delicate analysis to bound the "**Error**" term.

**Remark C.1.** *It should be pointed out that the decomposition on the terms "**First Order Main**", "**Error**" and "**Second Order Main**" originates from Wang et al. (2023). However, the power in the denominator of (Type II) is $\frac{1}{2} + \epsilon$, which is more complicated than the case $\frac{1}{2}$ used in Wang et al. (2023). The complicated power forces us to make a delicate analysis for (Type II).*

Before bounding the "**Error Term**", for the convenience of notation, we let $q_t = b + \sum_{i=1}^{t} \|g_i\|^{\gamma}$. Then by a direct computation, we can get

$$|\mathbb{E}_t[\langle \nabla f(x_t), (\eta_{t-1} - \eta_t) g_t \rangle]|$$

$$= \left| \mathbb{E}_t \left[ \left\langle \nabla f(x_t), \left( \frac{1}{(q_{t-1})^{\frac{1}{2}+\epsilon}} - \frac{1}{(q_t)^{\frac{1}{2}+\epsilon}} \right) g_t \right\rangle \right] \right|$$

$$= \left| \mathbb{E}_t \left[ \left\langle \nabla f(x_t), \left( \frac{(q_t)^{\frac{1}{2}+\epsilon} - (q_{t-1})^{\frac{1}{2}+\epsilon}}{(q_{t-1})^{\frac{1}{2}+\epsilon} (q_t)^{\frac{1}{2}+\epsilon}} \right) g_t \right\rangle \right] \right| \tag{30}$$

$$= \left| \mathbb{E}_t \left[ \left\langle \nabla f(x_t), \frac{(q_t)^{1+2\epsilon} - (q_{t-1})^{1+2\epsilon}}{(q_{t-1})^{\frac{1}{2}+\epsilon} (q_t)^{\frac{1}{2}+\epsilon} \left[ (q_t)^{\frac{1}{2}+\epsilon} + (q_{t-1})^{\frac{1}{2}+\epsilon} \right]} g_t \right\rangle \right] \right|$$

$$\le \mathbb{E}_t \left[ \|\nabla f(x_t)\| \frac{(q_t)^{1+2\epsilon} - (q_{t-1})^{1+2\epsilon}}{(q_{t-1})^{\frac{1}{2}+\epsilon} (q_t)^{\frac{1}{2}+\epsilon} \left[ (q_t)^{\frac{1}{2}+\epsilon} + (q_{t-1})^{\frac{1}{2}+\epsilon} \right]} \|g_t\| \right],$$

where we use Cauchy-Schwarz inequality in the first inequality.

Then we use the mean value theorem for the function $x^{1+2\epsilon}$ to get

$$\mathbb{E}_t\left[\|\nabla f\left(x_t\right)\|\,\frac{\left(q_t\right)^{1+2\epsilon}-\left(q_{t-1}\right)^{1+2\epsilon}}{\left(q_{t-1}\right)^{\frac{1}{2}+\epsilon}\left(q_t\right)^{\frac{1}{2}+\epsilon}\left[\left(q_t\right)^{\frac{1}{2}+\epsilon}+\left(q_{t-1}\right)^{\frac{1}{2}+\epsilon}\right]}\,\|g_t\|\right]$$

$$\leq \mathbb{E}_t\left[\|\nabla f\left(x_t\right)\|\,\frac{\left(1+2\epsilon\right)\left(q_t\right)^{2\epsilon}\|g_t\|^2}{\left(q_{t-1}\right)^{\frac{1}{2}+\epsilon}\left(q_t\right)^{\frac{1}{2}+\epsilon}\left[\left(q_t\right)^{\frac{1}{2}+\epsilon}+\left(q_{t-1}\right)^{\frac{1}{2}+\epsilon}\right]}\,\|g_t\|\right] \tag{31}$$

$$= \frac{\|\nabla f(x_t)\|}{\left(q_{t-1}\right)^{\frac{1}{2}+\epsilon}}\mathbb{E}_t\left[\frac{\left(1+2\epsilon\right)\left(q_t\right)^{\epsilon}\|g_t\|^2}{\left[\left(q_t\right)^{\frac{1}{2}+\epsilon}+\left(q_{t-1}\right)^{\frac{1}{2}+\epsilon}\right]}\,\frac{\|g_t\|}{\left(q_t\right)^{\frac{1}{2}}}\right],$$

Next we use the fact that $\|g_t\| \leq \left(q_t\right)^{\frac{1}{2}}$ (since $\|g_t\| \leq Q \leq b^{\frac{1}{2}} \leq \left(q_t\right)^{\frac{1}{2}}$) to obtain

$$\mathbb{E}_t\left[\|\nabla f\left(x_t\right)\|\,\frac{\left(q_t\right)^{1+2\epsilon}-\left(q_{t-1}\right)^{1+2\epsilon}}{\left(q_{t-1}\right)^{\frac{1}{2}+\epsilon}\left(q_t\right)^{\frac{1}{2}+\epsilon}\left[\left(q_t\right)^{\frac{1}{2}+\epsilon}+\left(q_{t-1}\right)^{\frac{1}{2}+\epsilon}\right]}\,\|g_t\|\right]$$

$$\leq \frac{\|\nabla f(x_t)\|}{\left(q_{t-1}\right)^{\frac{1}{2}+\epsilon}}\mathbb{E}_t\left[\frac{\left(1+2\epsilon\right)\left(q_t\right)^{\epsilon}\|g_t\|^2}{\left(q_t\right)^{\frac{1}{2}+\epsilon}+\left(q_{t-1}\right)^{\frac{1}{2}+\epsilon}}\right] \tag{32}$$

$$\leq \frac{\|\nabla f(x_t)\|}{\left(q_{t-1}\right)^{\frac{1}{2}+\epsilon}}\mathbb{E}_t\left[\frac{\left(1+2\epsilon\right)\left(q_t\right)^{\epsilon}\|g_t\|^2}{2\left(q_{t-1}\right)^{\frac{1}{2}+\epsilon}}\right],$$

where we use $q_t \geq q_{t-1}$ in the second inequality.

Simple algebra yields

$$\mathbb{E}_t\left[\|\nabla f\left(x_t\right)\|\,\frac{\left(q_t\right)^{1+2\epsilon}-\left(q_{t-1}\right)^{1+2\epsilon}}{\left(q_{t-1}\right)^{\frac{1}{2}+\epsilon}\left(q_t\right)^{\frac{1}{2}+\epsilon}\left[\left(q_t\right)^{\frac{1}{2}+\epsilon}+\left(q_{t-1}\right)^{\frac{1}{2}+\epsilon}\right]}\,\|g_t\|\right]$$

$$\leq \frac{\|\nabla f(x_t)\|}{\left(q_{t-1}\right)^{\frac{1}{2}+\epsilon}}\mathbb{E}_t\left[\frac{\left(1+2\epsilon\right)\left(q_t\right)^{\epsilon}\|g_t\|^2}{2\left(q_{t-1}\right)^{\frac{1}{2}+\epsilon}}\right]$$

$$= \frac{\|\nabla f(x_t)\|\left(q_t\right)^{\frac{\epsilon}{2}}}{\left(q_{t-1}\right)^{\frac{1}{2}+\epsilon}}\mathbb{E}_t\left[\frac{\left(1+2\epsilon\right)\left(q_t\right)^{\frac{\epsilon}{2}}\|g_t\|^2}{2\left(q_{t-1}\right)^{\frac{1}{2}+\epsilon}}\right]$$

$$\leq 2^{\frac{\epsilon}{2(1+2\epsilon)}}\frac{\|\nabla f(x_t)\|\left(q_{t-1}\right)^{\frac{\epsilon}{2}}}{\left(q_{t-1}\right)^{\frac{1}{2}+\epsilon}}\mathbb{E}_t\left[\frac{\sqrt{2}\left(q_t\right)^{\frac{\epsilon}{2}}\|g_t\|^2}{\left(q_t\right)^{\frac{1}{2}+\epsilon}}\right] \tag{33}$$

$$\leq 2^{\frac{1+3\epsilon}{2(1+2\epsilon)}}\frac{\|\nabla f(x_t)\|}{\left(q_{t-1}\right)^{\frac{1}{2}+\frac{\epsilon}{2}}}\mathbb{E}_t\left[\frac{Q\|g_t\|}{\left(q_t\right)^{\frac{1}{2}+\frac{\epsilon}{2}}}\right]$$

$$\leq 2^{\frac{1+3\epsilon}{2(1+2\epsilon)}}\left[\frac{1}{2}\frac{\|\nabla f(x_t)\|^2}{\left(q_{t-1}\right)^{1+\epsilon}}+\frac{1}{2}\left[\mathbb{E}_t\left(\frac{Q\|g_t\|}{\left(q_t\right)^{\frac{1}{2}+\frac{\epsilon}{2}}}\right)\right]^2\right]$$

$$\leq 2^{\frac{1+3\epsilon}{2(1+2\epsilon)}}\left[\frac{1}{2}\frac{\|\nabla f(x_t)\|^2}{\left(q_{t-1}\right)^{1+\epsilon}}+\frac{1}{2}\mathbb{E}_t\left(\frac{Q^2\|g_t\|^2}{\left(q_t\right)^{1+\epsilon}}\right)\right]$$

$$= \left(\frac{1}{2}\right)^{\frac{1+\epsilon}{2(1+2\epsilon)}}\left[\frac{\|\nabla f(x_t)\|^2}{\left(q_{t-1}\right)^{1+\epsilon}}+\mathbb{E}_t\left(\frac{Q^2\|g_t\|^2}{\left(q_t\right)^{1+\epsilon}}\right)\right],$$

where we use the proven fact $\left(q_{t-1}\right)^{1+2\epsilon} \geq \frac{1}{2}\left(q_t\right)^{1+2\epsilon}$ in (18) and $1+2\epsilon \leq 2$ in the second inequality; we use the fact $ab \leq \frac{1}{2}(a^2+b^2)$ in the fourth inequality; we use the fact $[\mathbb{E}_t(X)]^2 \leq \mathbb{E}_t(X^2)$ in the fifth inequality.

Putting (29), (30) and (33) together, we can obtain

$$\mathbb{E}_t \left[ f(x_{t+1}) \right] \le f(x_t) - \left[ 1 - \left( \frac{1}{2} \right)^{\frac{1+\epsilon}{2(1+2\epsilon)}} \right] \mathbb{E}_t \left[ \eta_{t-1} \| \nabla f(x_t) \|^2 \right]$$

$$+ \frac{M}{2} \mathbb{E}_t \left[ \frac{\| g_t \|^2}{\left( b + \sum_{i=1}^t \| g_i \|^\gamma \right)^{1+2\epsilon}} \right] + \left( \frac{1}{2} \right)^{\frac{1+\epsilon}{2(1+2\epsilon)}} Q^2 \, \mathbb{E}_t \left[ \frac{\| g_t \|^2}{\left( b + \sum_{i=1}^t \| g_i \|^\gamma \right)^{1+\epsilon}} \right]. \tag{34}$$

$\square$

*Proof of Theorem 3.7.* The proof of Theorem 3.7 is similar to Theorem 3.3, only exists a minor difference on coefficients. Concretely, in analogy of (17), we replace (2) with (3) to get

$$\mathbb{E}_t \left[ (t+1)^{1-\eta} \left( f(x_{t+1}) - f^* \right) \right]$$

$$\le t^{1-\eta} \left( f(x_t) - f^* \right) - \frac{D_4}{t^{\frac{1}{2}+\epsilon}} t^{1-\eta} \left( f(x_t) - f^* \right) + \frac{M}{2} \mathbb{E}_t \left[ \frac{(t+1)^{1-\eta} \| g_t \|^2}{\left( b + \sum_{i=1}^t \| g_i \|^\gamma \right)^{1+2\epsilon}} \right] \tag{35}$$

$$+ \left( \frac{1}{2} \right)^{\frac{1+\epsilon}{2(1+2\epsilon)}} Q^2 \, \mathbb{E}_t \left[ \frac{(t+1)^{1-\eta} \| g_t \|^2}{\left( b + \sum_{i=1}^t \| g_i \|^\gamma \right)^{1+\epsilon}} \right],$$

where $D_4 > 0$ is a constant. Actually we can directly get

$$\sum_{t=1}^\infty \mathbb{E}_t \left[ \frac{(t+1)^{1-\eta} \| g_t \|^2}{\left( b + \sum_{i=1}^t \| g_i \|^\gamma \right)^{1+2\epsilon}} \right] < \infty \quad \text{and} \quad \sum_{t=1}^\infty \mathbb{E}_t \left[ \frac{(t+1)^{1-\eta} \| g_t \|^2}{\left( b + \sum_{i=1}^t \| g_i \|^\gamma \right)^{1+\epsilon}} \right] < \infty$$

by Proposition 3.1. The remainder of the proof can follow the proof of Theorem 3.3. $\square$

*Proof of Theorem 3.8.* Replacing $1 - \eta = \frac{1}{2} - \epsilon$ in Proposition 3.1, we can get

$$\sum_{t=1}^\infty \mathbb{E}_t \left[ \frac{(t+1)^{\frac{1}{2}-\epsilon} \| g_t \|^2}{\left( b + \sum_{i=1}^t \| g_i \|^\gamma \right)^{1+2\epsilon}} \right] < \infty \quad \text{and} \quad \sum_{t=1}^\infty \mathbb{E}_t \left[ \frac{(t+1)^{\frac{1}{2}-\epsilon} \| g_t \|^2}{\left( b + \sum_{i=1}^t \| g_i \|^\gamma \right)^{1+\epsilon}} \right] < \infty.$$

The proofs can entirely follow the proof of Theorem 3.4. $\square$

*Proof of Theorem 3.9.* Note that for (Type II), we can directly get $\sum_{t=1}^\infty \mathbb{E}_t \left[ \eta_t^2 \| g_t \|^2 \right] < \infty$. The proofs can entirely follow the proof of Theorem 3.5. $\square$

Before providing the proof of Proposition 3.11, we present the following lemma. This lemma is an similar analogy to Lemma 1 in Orabona (2020a).

**Lemma C.2.** *Let $\{b_t\}$ and $\{a_t\}$ be two nonnegative sequences and $\{w_t\}$ be a sequence of vectors. Assume $\sum_{t=1}^\infty a_t b_t^p < \infty$ and $\sum_{t=1}^\infty a_t = \infty$, where $p \ge 1$. Furthermore, assume that there exists $L > 0$ such that*

$$|b_{t+\tau} - b_t| \le L \left( \sum_{i=t}^{t+\tau-1} a_i b_i + \left\| \sum_{i=t}^{t+\tau-1} w_i \right\| \right),$$

*where $w_t$ is such that $\| \sum_{t=1}^\infty w_t \|$ converges. Then $b_t$ converges to 0.*

*Proof.* Because $\sum_{t=1}^\infty a_t b_t^p < \infty$ and $\sum_{t=1}^\infty a_t = \infty$, we have $\liminf_t b_t = 0$. Then we only need to prove that $\limsup_t b_t = 0$. We will prove this by contradiction and let $\limsup_t b_t = \lambda > 0$. We discuss for two cases: $\lambda < \infty$ and $\lambda = \infty$. Firstly, assume $\lambda < \infty$.

As $\liminf_t b_t = 0$ and $\limsup_t b_t = \lambda$, there exists two sequences $\{m_j\}$ and $\{n_j\}$ such that

- $m_j < n_j < m_{j+1}$;

- $b_k > \frac{\lambda}{3}$ for $m_j \leq k < n_j$;

- $b_k \leq \frac{\lambda}{3}$ for $n_j \leq k < n_{j+1}$.

The convergence of the series implies that the sequence of partial sums are Cauchy sequences. Given that $\sum_{t=1}^{\infty} a_t b_t^p$ and $\|\sum_{t=1}^{\infty} w_t\|$ converge. Therefore, $\forall \epsilon > 0$, there exists $\hat{j}$ large enough such that for all $N \geq m_{\hat{j}}$, we have

$$\left\| \sum_{t=m_{\hat{j}}}^{N} a_t b_t^p \right\| \leq \epsilon \text{ and } \left\| \sum_{t=m_{\hat{j}}}^{N} w_t \right\| \leq \epsilon.$$

Set

$$\epsilon = \frac{\lambda}{6L} \min\{\frac{\lambda^{p-1}}{3^{p-1}}, 1\}.$$

Then for all $j \geq \hat{j}$ and all $m$ with $m_j \leq m < n_j$, we have

$$\begin{aligned}
\|b_{n_j} - b_m\| &\leq L \left( \sum_{i=m}^{n_j-1} a_i b_i + \left\| \sum_{i=m}^{n_j-1} a_i w_i \right\| \right) \\
&= \frac{3^{p-1}L}{\lambda^{p-1}} \sum_{i=m}^{n_j-1} a_i b_i \frac{\lambda^{p-1}}{3^{p-1}} + L \left\| \sum_{i=m}^{n_j-1} a_i w_i \right\| \\
&\leq \frac{3^{p-1}L}{\lambda^{p-1}} \sum_{i=m}^{n_j-1} a_i b_i^p + L \left\| \sum_{i=m}^{n_j-1} a_i w_i \right\| \\
&\leq \frac{3^{p-1}L}{\lambda^{p-1}} \epsilon + \epsilon L \\
&\leq \frac{3^{p-1}L}{\lambda^{p-1}} \frac{\lambda}{6L} \frac{\lambda^{p-1}}{3^{p-1}} + L\frac{\lambda}{6L} \\
&= \frac{\lambda}{3}.
\end{aligned} \tag{36}$$

Then

$$\|b_m\| = \|b_m - b_{n_j} + b_{n_j}\| \leq \|b_m - b_{n_j}\| + \|b_{n_j}\| \leq \frac{2\lambda}{3}.$$

This means for all $m \geq m_{\hat{j}}$, we have $b_m \leq \frac{2\lambda}{3}$, which contradicts $\limsup_t b_t = \lambda > 0$.

For the case $\lambda = \infty$, we can make the same argument to prove it. We complete the proof. $\square$

Next, we provide the proof of Proposition 3.11.

*Proof of Proposition 3.11.* For any $T$, summing over $t = 1$ to $T$ for (34), we have

$$\begin{aligned}
&\left[ 1 - \left(\frac{1}{2}\right)^{\frac{1+\epsilon}{2(1+2\epsilon)}} \right] \mathbb{E}_t \left[ \sum_{t=1}^{T} \eta_{t-1} \|\nabla f(x_t)\|^2 \right] \\
&\leq f(x_1) - f^* + \frac{M}{2} \mathbb{E}_t \left[ \sum_{t=1}^{T} \frac{\|g_t\|^2}{\left( b + \sum_{i=1}^{t} \|g_i\|^\gamma \right)^{1+2\epsilon}} \right] \\
&\quad + \left(\frac{1}{2}\right)^{\frac{1+\epsilon}{2(1+2\epsilon)}} Q^2 \mathbb{E}_t \left[ \sum_{t=1}^{T} \frac{\|g_t\|^2}{\left( b + \sum_{i=1}^{t} \|g_i\|^\gamma \right)^{1+\epsilon}} \right].
\end{aligned} \tag{37}$$

Taking the limit for $T \to \infty$, we can easily get

$$\left[1 - \left(\frac{1}{2}\right)^{\frac{1+\epsilon}{2(1+2\epsilon)}}\right] \mathbb{E}_t \left[\sum_{t=1}^{\infty} \eta_{t-1} \|\nabla f(x_t)\|^2\right] \leq \frac{M}{2} \mathbb{E}_t \left[\sum_{t=1}^{\infty} \frac{\|g_t\|^2}{\left(b + \sum_{i=1}^{t} \|g_i\|^\gamma\right)^{1+2\epsilon}}\right]$$

$$+ \left(\frac{1}{2}\right)^{\frac{1+\epsilon}{2(1+2\epsilon)}} Q^2 \mathbb{E}_t \left[\sum_{t=1}^{\infty} \frac{\|g_t\|^2}{\left(b + \sum_{i=1}^{t} \|g_i\|^\gamma\right)^{1+\epsilon}}\right] + f(x_1) - f^*. \tag{38}$$

From Lemma B.2, we have

$$\sum_{t=1}^{\infty} \frac{\|g_t\|^2}{\left(b + \sum_{i=1}^{t} \|g_i\|^\gamma\right)^{1+2\epsilon}} = \sum_{t=1}^{\infty} \frac{\|g_t\|^2 \|g_i\|^{2-\gamma}}{\left(b + \sum_{i=1}^{t} \|g_i\|^\gamma\right)^{1+2\epsilon}}$$

$$\leq \sum_{t=1}^{\infty} \frac{\|g_t\|^\gamma Q^{2-\gamma}}{\left(b + \sum_{i=1}^{t} \|g_i\|^\gamma\right)^{1+2\epsilon}} \tag{39}$$

$$< \infty.$$

and

$$\sum_{t=1}^{\infty} \frac{\|g_t\|^2}{\left(b + \sum_{i=1}^{t} \|g_i\|^\gamma\right)^{1+\epsilon}} = \sum_{t=1}^{\infty} \frac{\|g_t\|^2 \|g_i\|^{2-\gamma}}{\left(b + \sum_{i=1}^{t} \|g_i\|^\gamma\right)^{1+\epsilon}}$$

$$\leq \sum_{t=1}^{\infty} \frac{\|g_t\|^\gamma Q^{2-\gamma}}{\left(b + \sum_{i=1}^{t} \|g_i\|^\gamma\right)^{1+\epsilon}}$$

$$< \infty.$$

Then we have

$$\mathbb{E}_t \left[\sum_{t=1}^{\infty} \eta_{t-1} \|\nabla f(x_t)\|^2\right] < \infty. \tag{40}$$

Therefore,

$$\sum_{t=1}^{\infty} \eta_{t-1} \|\nabla f(x_t)\|^2 < \infty$$

almost surely. (We use a fact that a nonnegative variable $X$ with $\mathbb{E}[X] < \infty$ must satisfy $X < \infty$ almost surely.)

Also note that

$$\sum_{t=1}^{\infty} \eta_{t-1} \geq \sum_{t=1}^{\infty} \frac{1}{(b + (t-1)Q^\gamma)^{\frac{1}{2}+\epsilon}} = \infty.$$

Observe that

$$\|\|\nabla f(x_{t+\tau})\| - \|\nabla f(x_t)\|\| \leq \|\nabla f(x_{t+\tau}) - \nabla f(x_t)\|$$

$$= M \|x_{t+\tau} - x_t\|$$

$$= M \left\|\sum_{i=t}^{t+\tau-1} \eta_i g_i\right\| \tag{41}$$

$$= M \left\|\sum_{i=t}^{t+\tau-1} \eta_{i-1} \nabla f(x_i) + \sum_{i=t}^{t+\tau-1} \eta_i g_i - \sum_{i=t}^{t+\tau-1} \eta_{i-1} \nabla f(x_i)\right\|.$$

Let $w_t = \eta_t g_t - \eta_{t-1} \nabla f(x_t)$. To show $\|\sum_{t=1}^{\infty} w_t\|$ converges almost surely, we rewrite

$$\left\|\sum_{t=1}^{\infty} w_t\right\| = \left\|\sum_{t=1}^{\infty} \eta_{t-1} (g_t - \nabla f(x_t)) + \sum_{t=1}^{\infty} (\eta_t - \eta_{t-1}) g_t\right\|$$

$$\leq \left\|\sum_{t=1}^{\infty} \eta_{t-1} (g_t - \nabla f(x_t))\right\| + \left\|\sum_{t=1}^{\infty} (\eta_t - \eta_{t-1}) g_t\right\|. \tag{42}$$

Firstly, we prove

$$\left\| \sum_{t=1}^{\infty} \left( \eta_t - \eta_{t-1} \right) g_t \right\| < \infty.$$

This can be concluded by the following argument:

$$\left\| \sum_{t=1}^{\infty} \left( \eta_t - \eta_{t-1} \right) g_t \right\| \leq \sum_{t=1}^{\infty} \| \left( \eta_t - \eta_{t-1} \right) g_t \| \leq Q \sum_{t=1}^{\infty} \left( \eta_{t-1} - \eta_t \right) \leq Q \eta_0 < \infty.$$

Secondly, we prove almost surely,

$$\left\| \sum_{t=1}^{\infty} \eta_{t-1} \left( g_t - \nabla f(x_t) \right) \right\| < \infty.$$

Let $A_t = \sum_{i=1}^{t} \eta_{i-1} \left( g_i - \nabla f(x_i) \right)$. It is equivalent to prove $\lim_{t \to \infty} A_t$ exists almost surely. Since $\{g_i\}_{i=1}^{t}$ is unbiased and $g_i$ is not included in $\eta_{i-1}$, we can simply verify $A_t$ is a martingale. Therefore, by Theorem 12.1 in Williams (1991), to prove $\lim_{t \to \infty} A_t$ exists almost surely, it suffices to prove the martingale $A_t$ is bounded in $\mathcal{L}^2$. Also note that by Theorem 12.1 in Williams (1991), the martingale $A_t$ is bounded in $\mathcal{L}^2$ if and only if

$$\sum_{t=1}^{\infty} \mathbb{E}_t[\|A_t - A_{t-1}\|^2] < \infty.$$

Note that

$$
\begin{aligned}
\sum_{t=1}^{\infty} \mathbb{E}_t[\|A_t - A_{t-1}\|^2] &= \sum_{t=1}^{\infty} \mathbb{E}_t[\|\eta_{t-1} \left( g_t - \nabla f(x_t) \right)\|^2] \\
&\leq 2\mathbb{E}_t \left[ \sum_{t=1}^{\infty} \eta_{t-1}^2 \|g_t\|^2 \right] + 2\mathbb{E}_t \left[ \sum_{t=1}^{\infty} \eta_{t-1}^2 \|\nabla f(x_t)\|^2 \right] \\
&\leq 4\mathbb{E}_t \left[ \sum_{t=1}^{\infty} \eta_t^2 \|g_t\|^2 \right] + 2\mathbb{E}_t \left[ \sum_{t=1}^{\infty} \eta_{t-1}^2 \|\nabla f(x_t)\|^2 \right] \\
&\leq 4\mathbb{E}_t \left[ \sum_{t=1}^{\infty} \eta_t^2 \|g_t\|^2 \right] + 2\eta_1 \mathbb{E}_t \left[ \sum_{t=1}^{\infty} \eta_{t-1} \|\nabla f(x_t)\|^2 \right],
\end{aligned}
\tag{43}
$$

where we use the fact $\|a + b\|^2 \leq 2(\|a\|^2 + \|b\|^2)$ in the first inequality; we use the fact $\eta_{t-1}^2 \leq 2\eta_t^2$ in the second inequality (This can be proved similar to (18).); we use the fact $\eta_{t-1} \leq \eta_1$ in the third inequality.

By the above proven fact $\mathbb{E}_t \left[ \sum_{t=1}^{\infty} \eta_t^2 \|g_t\|^2 \right] < \infty$ in (39) and $\mathbb{E}_t \left[ \sum_{t=1}^{\infty} \eta_{t-1} \|\nabla f(x_t)\|^2 \right]$ in (40), we have

$$\sum_{t=1}^{\infty} \mathbb{E}_t[\|A_t - A_{t-1}\|^2] < \infty.$$

Combing all the above, we have proven that $\|\sum_{t=1}^{\infty} w_t\|$ converges almost surely.

Finally, applying Lemma C.2 with

$$w_t = \eta_t g_t - \eta_{t-1} \nabla f(x_t), \;\; a_t = \eta_{t-1}, \;\; b_t = \|\nabla f(x_t)\|, \;\; p = 2,$$

we have

$$\lim_{t \to \infty} \|\nabla f(x_t)\| = 0$$

almost surely. □

## D   DERIVATION STATED IN THE INSTRUCTION OF TABLE 1

Attia & Koren (2023) provides the high probability bounds for AdaGrad in both convex and nonconvex settings, which is the closest work to obtaining almost sure convergence rates of AdaGrad. Concretely, they prove that

(1) When $f(x)$ is convex, for any $\delta \in (0, \frac{1}{4})$, it holds with probability $1 - \delta$ that

$$f(\frac{1}{T} \sum_{t=1}^{T} x_t) - f^* = O\left( \frac{\sqrt{\log \frac{1}{\delta}}}{\sqrt{T}} \right). \tag{44}$$

(2) When $f(x)$ is nonconvex, for any $\delta \in (0, \frac{1}{3})$, it holds with probability $1 - \delta$ that

$$\frac{1}{T} \sum_{t=1}^{T} \|\nabla f(x_t)\|^2 = O\left( \frac{\log^2 \frac{T}{\delta}}{\sqrt{T}} \right). \tag{45}$$

Derivation stated in the instruction of Table 1 can be summarized in Corollary D.1. Corollary D.1 state that the high probability bound (with $\mathbf{polylog}(\frac{1}{\delta})$ term) can **partially** imply the almost sure convergence rates of AdaGrad.[2] The word "**partially**" originates that the almost sure convergence rates in Corollary D.1 are only valid for undetermined $t \geq t_0$ and in the sense of average-iterate for the convex case, while our results are valid for any time $t \geq 1$ and provide last-iterate convergence rates for the convex case.

**Corollary D.1.** *Consider the algorithm AdaGrad. We have*

*(1) For convex $f(x)$, there exists **unknown** $t_0$, for all $t \geq t_0$, it holds almost surely that*

$$f\left( \frac{1}{t} \sum_{t=1}^{t} x_i \right) - f^* = O\left( \frac{\sqrt{\log t}}{\sqrt{t}} \right). \tag{46}$$

*(2) For nonconvex $f(x)$, there exists **unknown** $t_0$, for all $t \geq t_0$, it holds almost surely that*

$$\min_{1 \leq i \leq t} \|\nabla f(x_i)\|^2 \leq \frac{1}{t} \sum_{i=1}^{t} \|\nabla f(x_i)\|^2 = O\left( \frac{\log^2 t}{\sqrt{t}} \right). \tag{47}$$

*Proof.* We provide the detailed proof for convex case. The nonconvex case follows the similar argument. For all $t$, define

$$E_t = \left\{ f\left( \frac{1}{t} \sum_{i=1}^{t} x_i \right) - f(x^*) = O\left( \sqrt{\frac{\log t}{t}} \right) \right\}$$

and $E_t^c$ by its complement. Replacing $\delta$ with $\delta_t = \frac{0.5}{t^2}$ into (44), one has

$$P(E_t) \geq 1 - \delta_t,$$

i.e.,

$$P(E_t^c) \leq \delta_t,$$

therefore

$$\sum_{t=1}^{\infty} P(E_t^c) \leq \sum_{t=1}^{\infty} \delta_t < \infty.$$

By Borel-Cantelli Lemma (See Lemma B.5 in Appendix B), one has

$$P(\limsup_{t \to \infty} E_t^c) = P(\cap_{t=1}^{\infty} \cup_{k=t}^{\infty} E_k^c) = 0,$$

i.e.,

$$P\left[ (\cap_{t=1}^{\infty} \cup_{k=t}^{\infty} E_k^c)^c \right] = P\left[ (\cup_{t_0=1}^{\infty} \cap_{t=t_0}^{\infty} E_t) \right] = 1,$$

which means there exists **unknown** $t_0$, for all $t \geq t_0$ such that

$$f\left( \frac{1}{t} \sum_{i=1}^{t} x_i \right) - f(x^*) = O\left( \sqrt{\frac{\log t}{t}} \right)$$

with probability 1. $\qquad \square$

---

[2]Corollary D.1 was implicit in Attia & Koren (2023), we explicitly state here.

