# OpenReview forum: "AdaGrad Converges in a Robust Sense: Almost Sure Last-Iterate Rates under Any Stopping Time"
_ICLR.cc/2026/Conference — Submitted to ICLR 2026_

### Official Review · Reviewer_CVdB · 2025-10-16

**Soundness:** 1
**Presentation:** 2
**Contribution:** 1
**Rating:** 2
**Confidence:** 4

**Summary:**

This paper studies the almost sure convergence of the AdaGrad-Norm-Type stepsize. Under certain assumptions, the authors prove the last-iterate convergence for (strongly) convex functions and best-iterate convergence for non-convex functions. All of the results hold for any time.

**Strengths:**

The paper is written clearly.

**Weaknesses:**

**Main Points**

1. First of all, the most critical issue is that none of the convergence results is proved for the original AdaGrad (or even AdaGrad-Norm).

    a) The Type I stepsize can never recover AdaGrad-Norm.

    b) For the Type II stepsize, none of the results allows $\epsilon=0$.

    Therefore, no results in this work can be applied to AdaGrad (or AdaGrad-Norm) directly.

1. The assumptions considered in the work are strong (in particular, for Assumption 2.2) and not even compatible. More concretely:

    a) Assumption 2.2 is a strong condition in general, especially under the smoothness condition (Assumption 1). Note that these two assumptions cannot hold together even for quadratic optimization on $\mathbb{R}^d$, since assumption 2.2 implies the true gradient is bounded.

    b) More fundamentally, it is known that Assumption 2.2 (in fact, its weaker version, i.e., $\nabla f(x)$ is bounded) and strong convexity cannot hold together on $\mathbb{R}^d$ (see, e.g., [1]). Therefore, Theorems 3.3 and 3.7 cannot be recognized as meaningful results.

    c) One of the key contributions in many prior works for AdaGrad/AdaGrad-Norm (or even more general adaptive gradient optimizers like Adam) is to prove the convergence without assuming bounded stochastic gradients (e.g., see [2, 3]). However, this study imposes it again.

    d) From a technical view, Assumption 2.2 can also reduce many difficulties in the proof.

1. For theoretical results and their proofs, there are also many inaccurate/incorrect places.

    a) The proof of Proposition 3.1 also seems incorrect. I don't see the point of $\sum_{t=1}^\infty\mathbb{E}\_t[\cdot]=\mathbb{E}\_t[\sum_{t=1}^\infty\cdot]$. If the authors keep the summation symbol outside. Then the current argument fails in many places. The same issue also appears in the proof of Proposition 3.11.

    b) Line 844, it should be up to $\xi_{t-1}$.

    c) In the proof, the term $a$ in the stepsize is missed in different places (e.g., inequality (30)).

    d) Line 1115, I cannot see why the equation holds.

1. For experiments, some improvements can also be made.

    a) The experiments are for stochastic optimization. Did the authors run multiple trials? If so, please report the number of trials and plot the error bar in the figure. Otherwise, the experiments are less convincing.

    b) Did the authors also run the Type-I stepsize? If yes, please also report the results. If not, then it confuses me why the paper studies it.

    c) The authors also set $\epsilon$ in the formulation of stepsize, but I cannot find it anywhere in the experiments section. What value of $\epsilon$ do the authors use? If the authors simply set it to $0$, does this mean introducing $\epsilon$ is only for the theoretical purpose?

    d) What is the value of $a$ in the experiments? Did the authors also tune it?

    e) The value of $b$ is also confusing. To the best of my knowledge, $b$ is usually a very small number in default, e.g., $10^{-10}$ to improve numerical stability. However, the value of $b$ here is kind of large. Did the authors tune it?

    f) In the caption of Figure 1, the authors wrote AdaGrad. Does this mean the authors in fact run AdaGrad instead of AdaGrad-Norm? Or it is just a typo.

**Other Points**

1. Line 245, there should be "(Robbins &Monro,1951)" for the citation (i.e., use \citep).

1. Line 256, missing a space after "... high probability.".

1. Line 275, if the authors don't formally define $g_t\equiv\nabla F(x_t,\xi_t)$, then I cannot see the purpose of requiring $f(x)\equiv\mathbb{E}[F(x,\xi)]$.

1. In Lemma B.3, the subscripts $t$ and $n$ should not both be used. Please fix this issue.

1. In Lemma B.5, it should be $\sum_{n}^\infty$.

1. In Corollary D.1 and its proof, many points can be further improved:

    a) The authors use both $f^\*$ and $f(x^*)$. I suggest keeping only $f^\*$ to make the notation clearer.

    b) $\sum_{i=1}^tx_t$ should be $\sum_{i=1}^tx_i$.

    c) The authors should use either $P[]$ or $P()$, but not both. In addition, the notation $P$ is not consistent with $\mathbb{P}$ in Lemma B.5.

    d) The statement about $t_0$ is also ambiguous, as $t_0$ should depend on the sample path, meaning that it may not be a uniform value.

**References**

[1] Nguyen, Lam, et al. "SGD and Hogwild! convergence without the bounded gradients assumption." International Conference on Machine Learning. PMLR, 2018.

[2] Faw, Matthew, et al. "The power of adaptivity in sgd: Self-tuning step sizes with unbounded gradients and affine variance." Conference on Learning Theory. PMLR, 2022.

[3] Wang, Bohan, et al. "Convergence of adagrad for non-convex objectives: Simple proofs and relaxed assumptions." The Thirty Sixth Annual Conference on Learning Theory. PMLR, 2023.

**Questions:**

Please refer to **Weaknesses** above.

**Details Of Ethics Concerns:**

NA.

---

### Official Review · Reviewer_1Afv · 2025-10-28

**Soundness:** 2
**Presentation:** 2
**Contribution:** 2
**Rating:** 2
**Confidence:** 4

**Summary:**

This paper studies the convergence of the step-size version of ADAGRAD for convex, strongly-convex and non-convex settings. It focuses on the almost-sure convergence of either function values, gradients or iterates (depending on the assumptions on $f$) in terms of so-called "last iterate". That is for example studying $f(x_t)$ and not $f(\bar{x}_t)$ where $\bar{x}_t$ is a weighted-average of past iterates. Previous work on this problem either gets convergence with high probability $1-\delta$ (where a.s. convergence is $\delta=0$), or convergence for averaged iterates. I share the motivation given by the author that last-iterate convergence gives a theoretical result closer to practice, therefore the problem studied is important given how often used Adagrad is in machine learning.
Unfortunately, the author do not study the usual step-size version of Adagrad, but do a slight modification by replacing the square exponent by a tunable parameter $\gamma\in (0,2)$. While this may seem harmless at first sight, the choices of $\gamma$ for which the results derived are satisfactory are those that almost anihilate the "Adagrad" style of the step-size and the method almost boils down to SGD (see below more details in the Weakness section). In short, the problem studied is interesting, but it seems to me that the paper achieves much less than what is claimed in the title, abstract and introduction. In the end, I do not think the results are strong enough for publication in a major conference.

**Strengths:**

* The problem studied is of interest to the machine learning community.
* The theoretical study is thorough, it covers the main cases of differentiable optimization: strongly-convex, convex and non-convex settings (although I do not think the results are very meaningful).
* Prior work is clearly discussed and cited.

**Weaknesses:**

Below I first list the three main issues with the paper (in my opinion). The rest are more minor concerns that can be fixed rather easily.
1. I think the title and abstract of the paper are really misleading, I understand this is likely not intentional but I encourage the authors to correct it. Indeed, it is only through a footnote on page 2 that we understand that this paper does not study the standard Adagrad but step-size Adagrad (or Norm-Adagrad). Later we realize that this is actually not exactly step-size Adagrad since the parameter $\gamma$ cannot be taken equal to $2$. Finally at the end the results derived are strong when $\gamma\simeq 0$ which makes the method significantly different from Adagrad (it is almost SGD with step-size $1/\sqrt{t}$, see hereafter).
2. To derive the theorems in Section 3, the authors make a uniform boundedness assumption on the gradients $\Vert g_t\Vert^2\leq Q$. While this is a strong assumptions it is indeed common.
The problem is that later in Section 3 the theorems contain a condition on $b$ (the constant term in the denominator of the step-size), and this condition depends on $Q$. But if we sumarize, $Q$ is a uniform bound on the stochastics gradients of the trajectory, so it depends on the sequence $(x_t)_t$. But the sequence depends itself on the choice of the step-size, so it depends on $b$. Therefore, $b$ depends on $Q$ which itselfs depends on $b$. That is a circular argument that I believe makes the result less meaninful in practice (because the condition on $b$ can never be checked a priori).
3. What I believe to be the main issue with the theoretical results is that the rates derived are stated as a function of $\eta$ or $\epsilon$: they have either the form  $1/t^{1-\eta}$ or $1/t^{1/2 - \epsilon}$. But this hides the fact that $\eta$ and $\epsilon$ depend on $\gamma$, indeed there are conditions on $\gamma$. For example in Theorem 3.4, the author give two conditions on $\epsilon$ that boil down to $\frac{\gamma}{4} \leq \epsilon\leq \frac{1}{2}$. So the rate obtained is of order $1/2 -\epsilon$ which is no better than $1/2 - \frac{\gamma}{4}$.
So to obtain a rate close to $1/2$ (the same as for SGD), we need $\gamma \simeq 0$, but in that case $\Vert g_t\Vert^\gamma \simeq 1$ (because $\Vert g_t\Vert^0=1$), so $\sum_i^t \Vert g_i\Vert^\gamma\simeq \sum_i^t 1 = t$. In other words, **when $\gamma\simeq 0$ the algorithm almost boils down to SGD with decreasing step-sizes**, for which those results are known. So the results are meaninful for $\gamma$ small, but then the FlexAdagrad algorithm proposed is not close to ADAGRAD anymore.

* I think the way the results are presented hides the issues above by giving less explicit conditions on $\epsilon$ and that this should really be clarified.

Minor concerns:
* The main interest of adaptive methods is to ease the tuning of the step-size parameter. I am afraid that adding an additional hyper-parameter $\gamma$ in Adagrad makes the tuning more complicated. Hence I am not sure that it really brings "flexibility" to Adagrad as stated
* In line with what is stated before, the authors discuss "last iterate convergence" without properly defining it. Last iterate convergence could be in terms of $x_t$, $f(x_t)$, or $\nabla f(x_t)$ and only late in the paper we see what types of results are proven.
* As the authors explain, assumptions 2.2 is key to deduce that the step-sizes $\eta_t$ are square-summable, which is key for deriving the results. Yet this condition cannot be checked a priori (as properly discussed in the paper). Proving the square-summability is essentially the main difficulty for methods with adaptive (hence random) step-sizes, and thus Assumption 2.2 circumvents the key difficulty when studying Adagrad.
* The parameter $\eta$ in the rates is already used for the step-size $\eta_t$, I suggest using another letter.

**Questions:**

* Could you please clarify the main issue with the dependence of $\eta$ and $\epsilon$ on $\gamma$ as discussed above, and how $\gamma$ affect the result?
* In the experiments of Figure 1, I suggests the author compare to SGD with descreasing step-sizes $1/\sqrt{k}$. I expect it to give results close to the curve for $\gamma=0.01$.
* In Figure 1, it seems that all the train losses are comparable, hence it is not clear that $\gamma$ has a strong effect. Did the author try other problems?
* What do the author mean by saying in introduction that there is need to provide "more flexibility in the implementation of Adagrad"? Has this been a problem reported before?
* Page 2: could the author clarify why rates in high-probability only offer finite-time horizon confidence unlike almost sure convergence?
* Page 3: the authors wrote "with loss of generality we can take $a=1$. I assume they meant *without*, but I actually do not believe one can take $a=1$ without loss of generality. If so, could they prove it?
* What does the condition on $b$ gives in practice? Because in vanilla implementations, $b$ is usually very small ($10^{-8}$), is it the case here with the theorems that are stated?

---

### Official Review · Reviewer_W1oe · 2025-10-31

**Soundness:** 1
**Presentation:** 2
**Contribution:** 2
**Rating:** 2
**Confidence:** 3

**Summary:**

In this paper, the authors study a variant of AdaGrad-Norm where the update exponents are hyperparameters denoted by $\gamma$ and $\epsilon$ and analyze its almost-sure convergence rates under strongly convex, convex, and non-convex settings, respectively. Specifically, for the strongly convex case, the authors claim an almost-sure convergence rate of $o(t^{-\frac{1}{2}+\frac{\gamma}{4}})$ for the last iterate. In the convex case, they establish a last-iterate rate of $O(t^{-1/2+\epsilon})$. Finally, for the non-convex setting, they show a rate of $O(t^{-1/2+\epsilon})$ in terms of the best iterate.

**Strengths:**

Most prior works on adaptive methods analyze convergence in expectation or with high probability, which does not guarantee convergence of individual sample trajectories. In contrast, this paper studies almost-sure convergence rates, providing a stronger and more robust guarantee. To the best of my knowledge, such results have only been previously established for SGD and its variants with predetermined diminishing step sizes. For the convex setting, the authors also claim to obtain an almost-sure rate of $O(t^{-\frac{1}{2}+\epsilon})$ for the last iterate, which improves upon the looser last-iterate rates known for SGD. However, I have concerns about the correctness of the results, as detailed in the next section.

**Weaknesses:**

- **Correctness of Proposition 3.1.** This is my main concern, since Proposition 3.1 is the cornerstone of the subsequent convergence results. I don't think the inequality in (1) is valid under the stated assumptions. As a counterexample, consider the case where $\\|g\_t\\| = Q$ for all $t \geq 1$. Then the sum in (1) becomes $\\sum\_{t=1}^{\infty} \frac{t^{1-\eta}Q^2}{(b+t \cdot Q^\gamma)^{1+\beta}}$, which,  in the limit $b\rightarrow 0$, further reduces to $\sum_{t=1}^{\infty} {t^{-\eta-\beta}Q^{2-\gamma(1+\beta)}}$. This series is only finite when $\eta+ \beta > 1$, which is stricter than the condition stated in the proposition. Looking at the proof of Proposition 3.1, I believe the error appears in the last step of Case (II). The authors claim that $\sum_{j=1}^{\infty} \frac{j^{1-\eta} \\|\tilde{g}_j\\|^\gamma}{(b+\sum\_{i=1}^j \\|\tilde{g}\_i\\|^\gamma)^{1+\beta}} < \infty$ follows from the result of Case (I). However, the bound proved in Case (I) does not include the extra factor $j^{1-\eta}$, so the stated implication does not go through.

- **Strength of the convergence results.** Assuming the aforementioned error can be corrected, I find the convergence results to be somewhat weak. First, in the strongly convex setting, the condition in Proposition 3.1 implies that $1-\eta \leq \frac{1}{2} - \frac{\gamma}{4}$. Consequently, the convergence rate cannot be better than $o(t^{-\frac{1}{2}+\frac{\gamma}{4}})$, which is slower than the corresponding rate achieved by SGD. Moreover, in both the strongly convex and convex cases, the established rate becomes vacuous when $\gamma = 2$, which corresponds to the standard AdaGrad, and the best rate is obtained when $\gamma$ approaches 0, which makes the step size less adaptive and more similar to a predetermined diminishing step size of $\frac{a}{(b+t)^{\frac{1}{2}+\epsilon}}$. Thus, the presented results do not convincingly demonstrate the advantage of AdaGrad over standard SGD.

**Questions:**

I am confused by the authors' claim that the almost sure results hold for any stopping time $t \geq 1$. In my understanding, the convergence results presented in this paper are asymptotic and are only valid when the number of iterations $t$ is sufficiently large. Therefore, they do not provide a non-asymptotic guarantee that holds uniformly for all $t$.

---

### Official Review · Reviewer_PYxr · 2025-11-01

**Soundness:** 3
**Presentation:** 3
**Contribution:** 3
**Rating:** 6
**Confidence:** 3

**Summary:**

This paper investigates the **almost sure (a.s.) convergence rates** of **AdaGrad**—one of the most widely used adaptive gradient methods—under both **convex and non-convex settings**, and crucially, for **the last-iterate** rather than the average-iterate.
The authors introduce a generalization called **FlexAdaGrad-Norm**, which includes a **flexibility parameter** ( \gamma \in [0,2] ) that controls the influence of past gradient norms in the adaptive step size.

They rigorously establish:

* **Last-iterate a.s. convergence rates** for strongly convex and convex cases.
* **Best-iterate a.s. convergence rates** for the non-convex case.
* All results hold **for any random stopping time ( t \ge 1 )**, strengthening robustness claims.

Experiments on CIFAR-10 using VGG+BN+Dropout empirically validate that **smaller ( \gamma )** values (e.g., 0.1 or 1) outperform the standard ( \gamma = 2 ), improving stability and accuracy.

**Strengths:**

1. **Significant Theoretical Contribution**

   * The paper closes an important gap by providing **last-iterate almost sure convergence rates** for AdaGrad, which had not been addressed before.
   * The theoretical results generalize previous works on SGD (e.g., Liu & Yuan, 2022; Sebbouh et al., 2021) and extend them to **adaptive methods** with **non-trivial step-size dependencies**.
   * The introduction of **generalized square summability** is an elegant technical innovation that bridges the step-size condition of SGD and AdaGrad.

2. **Comprehensive Coverage**

   * The analysis includes both **Type I** and **Type II** step-size definitions, encompassing all known AdaGrad variants.
   * Convergence results are derived for **strongly convex**, **convex**, and **non-convex** objectives.

3. **Robustness under Arbitrary Stopping Times**

   * The results hold for any stopping time, offering theoretical justification for adaptive stopping mechanisms (e.g., early stopping, validation-based termination).
   * This robustness is practically relevant in modern training pipelines.

4. **Clarity and Theoretical Soundness**

   * The proofs are systematic and well-structured, relying on classical martingale tools (Robbins–Siegmund theorem) and new bounding techniques.
   * The comparison table (Table 1) clearly positions the contribution relative to prior work.

5. **Empirical Validation of Flexibility Parameter**

   * Simple yet effective experiments demonstrate that introducing ( \gamma ) can improve empirical performance and training stability.

**Weaknesses:**

1. **Limited Experimental Depth**

   * The empirical section is minimal and focuses only on CIFAR-10 with a single architecture (VGG).
   * There are no large-scale or ablation studies exploring how ( \gamma ) interacts with learning rate, batch size, or non-convex architectures (e.g., transformers).

2. **Missing Intuition for Flexibility Parameter**

   * While ( \gamma ) is well-motivated mathematically, the **practical intuition**—why smaller values improve stability—is not sufficiently explained.
   * A discussion linking ( \gamma ) to adaptivity or implicit regularization would improve accessibility.

3. **Notation Density**

   * The paper introduces many parameters (e.g., ( \eta, \gamma, \epsilon, \beta, \mu, M, Q )), which makes the reading heavy.
   * A summarizing table of notation early in the paper would be very helpful.

4. **No Comparison to Other Adaptive Methods**

   * The work focuses solely on AdaGrad; even though the authors mention Adam/AdamW as future directions, no empirical or theoretical comparisons are made.
   * It is unclear whether similar results could hold for other adaptive algorithms.

5. **Minor Clarity Issues**

   * Some propositions and inequalities are overly detailed in the main text, pushing key insights into appendices.
   * Figures could use clearer legends and scaling (e.g., Figure 1’s axes).

**Questions:**

1. Can the “generalized square summability” condition be extended to **matrix-based AdaGrad** (e.g., full or diagonal preconditioning)?
2. How sensitive are the theoretical guarantees to the bounded-gradient assumption (Assumption 2.2)? Could relaxed moment bounds suffice?
3. For non-convex functions, is there an intuitive explanation for why **only best-iterate** convergence (and not last-iterate) is provable?
4. Could the framework be extended to stochastic adaptive momentum methods (e.g., Adam) without losing almost sure guarantees?

---

### Meta-Review · Area_Chair_Rzbm · 2025-12-10

**Summary:**

The reviewers have indicated several issues with this submission. Two reviewers indicate that the proof is not rigorous and there are some gaps in the analysis. Some reviewers mention that the assumptions are strong, for example, Assumption 2.2 is strong. Two reviewers also mention that the the results are not meaningful as it requires $\gamma\approx 0$ to get best rates. However, in this case, the proposed algorithm is not close to ADAGRAD anymore.

**Reviewer Concerns:**

The authors do not post responses. Therefore, all the concerns are still outstanding.

**Reviewer Scores:**

The reviewers would not change score since there are no responses from authors.

---

### Decision · Program_Chairs · 2026-01-26

Reject